# E. coli TraR allosterically regulates transcription initiation by altering RNA polymerase conformation

James Chen[1], Saumya Gopalkrishnan[2], Courtney Chiu[1], Albert Y Chen[2], Elizabeth A Campbell[1], Richard L Gourse[2], Wilma Ross[2], Seth A Darst[1]*

[1]The Rockefeller University, New York, United States; [2]Department of Bacteriology, University of Wisconsin-Madison, Madison, United States

**Abstract** TraR and its homolog DksA are bacterial proteins that regulate transcription initiation by binding directly to RNA polymerase (RNAP) rather than to promoter DNA. Effects of TraR mimic the combined effects of DksA and its cofactor ppGpp, but the structural basis for regulation by these factors remains unclear. Here, we use cryo-electron microscopy to determine structures of *Escherichia coli* RNAP, with or without TraR, and of an RNAP-promoter complex. TraR binding induced RNAP conformational changes not seen in previous crystallographic analyses, and a quantitative analysis revealed TraR-induced changes in RNAP conformational heterogeneity. These changes involve mobile regions of RNAP affecting promoter DNA interactions, including the βlobe, the clamp, the bridge helix, and several lineage-specific insertions. Using mutational approaches, we show that these structural changes, as well as effects on $\sigma^{70}$ region 1.1, are critical for transcription activation or inhibition, depending on the kinetic features of regulated promoters.

*For correspondence: darst@rockefeller.edu

Competing interests: The authors declare that no competing interests exist.

## Introduction

Transcription initiation is a major control point for gene expression. In bacteria, a single RNA polymerase (RNAP) performs all transcription. In *Escherichia coli* (*Eco*), the essential primary σ factor, $\sigma^{70}$, binds to RNAP to form the $\sigma^{70}$-holoenzyme ($E\sigma^{70}$) that is capable of recognizing and initiating at promoters for most genes. Upon locating the promoter, $E\sigma^{70}$ melts a ~ 13 bp segment of DNA to form the open promoter complex (RPo) in which the DNA template strand (t-strand) is loaded into the RNAP active site, exposing the transcription start site (*Bae et al., 2015b*; *Zuo and Steitz, 2015*). A key feature of the RPo formation pathway is that it is a multi-step process, with the RNAP-promoter complex passing through multiple intermediates before the final, transcription competent RPo is formed (*Hubin et al., 2017a*; *Ruff et al., 2015a*; *Saecker et al., 2002*).

A variety of transcription factors bind to the promoter DNA and/or to RNAP directly to regulate initiation (*Browning and Busby, 2016*; *Haugen et al., 2008*). Bacterial RNAP-binding factors, encoded by the chromosome or by bacteriophage or extrachromosomal elements, interact with different regions of the enzyme to regulate its functions (*Haugen et al., 2008*). One such factor is ppGpp, a modified nucleotide that functions together with the RNAP-binding protein DksA in *Eco* to reprogram bacterial metabolism in response to nutritional stresses during the so-called stringent response. Following amino acid starvation, ppGpp is synthesized by the RelA factor in response to uncharged tRNAs in the ribosomal A site (*Brown et al., 2016*; *Cashel and Gallant, 1969*; *Ryals et al., 1982*). Together, ppGpp and DksA alter the expression of as many as 750 genes within 5 min of ppGpp induction (*Paul et al., 2004a*; *Paul et al., 2005*; *Sanchez-Vazquez et al., 2019*), inhibiting, for example, promoters responsible for ribosome biogenesis and activating promoters responsible for amino acid synthesis.

**eLife digest** Cells need to make proteins in order to survive. To make a protein, a cell first needs to take the information from a gene coded in its DNA and copy it into a template made of a different molecule called RNA via a process called transcription. Not all genes are transcribed or expressed at the same time or the same rate, because a cell needs different proteins depending on its environment. The molecules that regulate when each gene is transcribed are called transcription factors.

When the bacterium *Escherichia coli* is starved for the building blocks it needs to make proteins, it changes the expression of almost one quarter of its genes within 5 minutes. This broad response requires two transcription factors, ppGpp and DksA. Unlike most transcription factors, these two molecules bind directly to the enzyme responsible for transcribing DNA into RNA, the RNA polymerase, but they do not bind to DNA.

Another transcription factor called TraR can mimic the combined effects of ppGpp and DksA on transcription, changing the conformation of RNA polymerase in the same way. Now, Chen et al. have used a high-resolution imaging technique called cryo-electron microscopy to reveal the details of how the structure of RNA polymerase changes in response to TraR binding, and by analogy, in response to ppGpp and DksA.

The experiments showed that TraR interacts with regions of RNA polymerase that move when the protein binds DNA. These interactions can either help or hinder the start of transcription depending on which DNA sequence the RNA polymerase binds. Using the structures obtained via cryo-electron microscopy as guides, Chen et al. mutated each of the mobile parts of RNA polymerase individually to determine which interactions with TraR were necessary to change gene expression.

These results shed light on a fundamental process in all living cells, the initiation of transcription, and how it changes in response to the cell's nutritional environment. This may help explain how different cells regulate gene expression and may also lead to the development of new antibiotics.

The overall RNAP structure is reminiscent of a crab claw, with one pincer comprising primarily the β′ subunit, and the other primarily the β subunit (*Zhang et al., 1999*). Between the two pincers is a large cleft that contains the active site. In Eσ$^{70}$ without nucleic acids, this channel is occupied by the σ$^{70}_{1.1}$ domain which is ejected upon entry of the downstream duplex DNA (*Bae et al., 2013*; *Mekler et al., 2002*). The Bridge Helix (BH) bridges the two pincers across the cleft, separating the cleft into the main channel, where σ$^{70}_{1.1}$ or nucleic acids reside, and the secondary channel, where NTPs enter the RNAP active site.

DksA binds in the RNAP secondary channel (*Lennon et al., 2012*; *Molodtsov et al., 2018*; *Perederina et al., 2004*). ppGpp binds directly to RNAP at two binding sites: site 1, located at the interface of the β′ and ω subunits (*Ross et al., 2013*; *Zuo et al., 2013*), and site 2, located at the interface of β′ and DksA (*Molodtsov et al., 2018*; *Ross et al., 2016*). The ppGpp bound at site one inhibits transcription ~2 fold under conditions where the effects of ppGpp bound at both sites together with DksA are as much as 20-fold (*Paul et al., 2004b*; *Ross et al., 2016*). By contrast, ppGpp/DksA at site two has a larger effect on inhibition and is necessary and sufficient for activation (*Ross et al., 2016*).

TraR is a distant homolog of DksA. Although only half the length of DksA, TraR regulates *Eco* transcription by binding to the RNAP secondary channel and mimicking the combined effects of ppGpp and DksA (*Gopalkrishnan et al., 2017*). TraR is encoded by the conjugative F plasmid and is expressed from the pY promoter as part of the major *tra* operon transcript (*Frost et al., 1994*; *Maneewannakul and Ippen-Ihler, 1993*). Like DksA, TraR inhibits Eσ$^{70}$-dependent transcription from ribosomal RNA promoters (e.g. *rrnB* P1) and ribosomal protein promoters (e.g. *rpsT* P2, expressing S20), and activates amino acid biosynthesis and transport promoters (e.g. p*thrABC*, p*hisG*, p*argI*, p*livJ*) in vivo and in vitro (*Gopalkrishnan et al., 2017*). The affinity of TraR for RNAP is only slightly higher than that of DksA, yet its effects on promoters negatively regulated by ppGpp/DksA in vitro are as large or larger than those of ppGpp/DksA (*Gopalkrishnan et al., 2017*). The effects of TraR on promoters positively regulated by ppGpp/DksA are also independent of ppGpp (*Gopalkrishnan et al., 2017*).

Models for ppGpp/DksA and TraR binding to RNAP have been proposed based on biochemical and genetic approaches (*Gopalkrishnan et al., 2017*; *Parshin et al., 2015*; *Ross et al., 2013*; *Ross et al., 2016*). Crystal structures of ppGpp/DksA/RNAP and TraR/RNAP confirmed the general features of these models and provided additional detail about their interactions with RNAP (*Molodtsov et al., 2018*), but did not reveal the mechanism of inhibition or activation, in large part because of crystal packing constraints on the movement of mobile regions of the complex. Thus, the structural basis for the effects of ppGpp/DksA or TraR on transcription has remained elusive.

To help understand TraR regulation and principles of the regulation of transcription initiation in general, we used single particle cryo-electron microscopy (cryo-EM) to examine structures of Eσ$^{70}$ alone, Eσ$^{70}$ bound to TraR (TraR-Eσ$^{70}$), and Eσ$^{70}$ bound to a promoter inhibited by TraR [*rpsT* P2; *Gopalkrishnan et al., 2017*]. Cryo-EM allows the visualization of multiple discrete conformational states populated in solution and in the absence of crystal packing constraints. Furthermore, new software tools allow for the analysis of continuous distributions of conformational heterogeneity in the cryo-EM data (*Nakane et al., 2018*).

The TraR-Eσ$^{70}$ structures show TraR binding in the secondary channel of the RNAP, consistent with the TraR-Eσ$^{70}$ model (*Gopalkrishnan et al., 2017*) and crystal structure (*Molodtsov et al., 2018*). However, the cryo-EM structures reveal major TraR-induced changes to the RNAP conformation that were not evident in the crystal structure due to crystal packing constraints. Structural analyses generated mechanistic hypotheses for TraR function in both activation and inhibition of transcription that were then tested biochemically. Based on the combined structural and functional analyses, we propose a model in which TraR accelerates multiple steps along the RPo formation pathway and at the same time modulates the relative stability of intermediates in the pathway. Whether a promoter is activated or inhibited by TraR is determined by the intrinsic kinetic properties of the promoter (*Galburt, 2018*; *Haugen et al., 2008*; *Paul et al., 2005*).

## Results

### Cryo-EM structures of TraR-Eσ$^{70}$, Eσ$^{70}$, and *rpsT* P2 RPo

We used single-particle cryo-EM to examine the structure of the *Eco* TraR-Eσ$^{70}$ complex in the absence of crystal packing interactions that could restrict conformational states of the complex. We also determined cryo-EM structures of Eσ$^{70}$ alone and the Eσ$^{70}$-*rpsT* P2 promoter RPo for comparison. TraR function under cryo-EM solution conditions (*Chen et al., 2019*) was indistinguishable from function under standard in vitro transcription assay conditions (*Figure 1—figure supplement 1A–C*).

Analysis of the cryo-EM data for the TraR-Eσ$^{70}$ complex gave rise to three distinct conformational classes (*Figure 1—figure supplement 1D*). All three structures are essentially identical except for the disposition of Si3 [also called β′i6; *Lane and Darst, 2010a*], a 188-residue lineage-specific insertion (LSI) in the trigger-loop (TL) of *Eco* RNAP (*Chlenov et al., 2005*) (*Figure 1A,B*). The first class [TraR-Eσ$^{70}$(I)] contained approximately 41% of the particles and resolved to a nominal resolution of 3.7 Å (*Figure 1A*). The second class [TraR-Eσ$^{70}$(II)] contained approximately 33% of the particles and resolved to a nominal resolution of 3.8 Å (*Figure 1B*). The third class [TraR-Eσ$^{70}$(III)] contained the remaining 26% of the particles and resolved to a nominal resolution of 3.9 Å (*Figure 1—figure supplement 1D*, *Figure 1—figure supplement 2*; *Supplementary file 1*). With Si3 (β′ residues 948–1126) excluded, the structures superimpose with a root-mean-square deviation (rmsd) of 0.495 Å over 3,654 α-carbons.

The overall binding mode of TraR in the cryo-EM structures (*Figure 1A–D*) is consistent with the effects of TraR or RNAP substitutions on TraR function (*Gopalkrishnan et al., 2017*) and is broadly consistent with the X-ray structure (*Molodtsov et al., 2018*). TraR can be divided into three structural elements, an N-terminal helix (TraR$_N$, residues 2–27; *Figure 1D,E*), a globular domain (TraR$_G$, residues 28–57; *Figure 1D,F*), and a C-terminal helix (TraR$_C$, residues 58–73; *Figure 1D,G*). A 4-Cys Zn$^{2+}$-binding motif spans TraR$_G$ and TraR$_C$ (*Figure 1F*). TraR$_N$ extends from the RNAP active site out through the RNAP secondary channel to the β′rim-helices (at the entrance to the RNAP secondary channel), interacting with key RNAP structural elements surrounding the active site, including the -NADFDGD- motif that chelates the active site Mg$^{2+}$ (*Zhang et al., 1999*), the F-loop (*Miropolskaya et al., 2009*), and the bridge-helix (*Figure 1D*). The N-terminal tip of TraR$_N$ (TraR

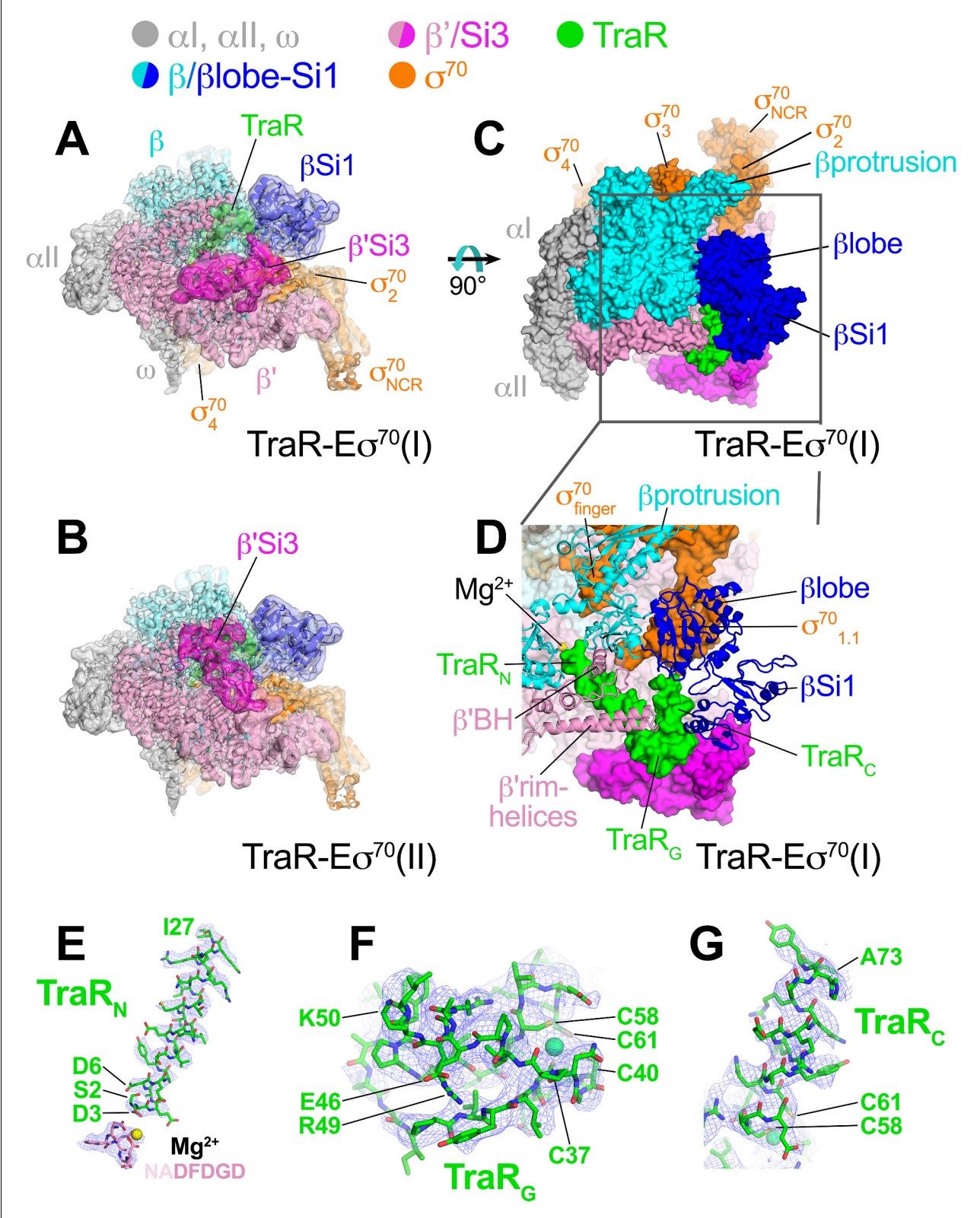

**Figure 1.** Cryo-EM structure of TraR-Eσ$^{70}$. (*top*) Color-coding key. (**A**) TraR-Eσ$^{70}$(I) - cryo-EM density map (3.7 Å nominal resolution, low-pass filtered to the local resolution) is shown as a transparent surface and colored according to the key. The final model is superimposed. (**B**) TraR-Eσ$^{70}$(II) - cryo-EM density map (3.8 Å nominal resolution, low-pass filtered to the local resolution) is shown as a transparent surface and colored according to the key. The final model is superimposed. (**C**) Top view of TraR-Eσ$^{70}$(I). The boxed area is magnified in (**D**). (**D**) Magnified top view of TraR-Eσ$^{70}$(I) - shows TraR$_N$

*Figure 1 continued on next page*

*Figure 1 continued*

(starting near RNAP active site Mg$^{2+}$, extending out secondary channel), TraR$_G$ (interacting primarily with β'rim-helices), and TraR$_C$ (interacting with βlobe-Si1). (**E – G**) Cryo-EM density (blue mesh) defining the TraR structure. (**E**) TraR$_N$ and -NADFDGD- motif of RNAP β' (chelating active site Mg$^{2+}$). (**F**) TraR$_G$. (**G**) TraR$_C$.

The online version of this article includes the following figure supplement(s) for figure 1:

**Figure supplement 1.** Cryo-EM solution conditions do not affect TraR function and TraR-Eσ$^{70}$ cryo-EM processing pipeline.
**Figure supplement 2.** TraR-Eσ$^{70}$ cryo-EM.
**Figure supplement 3.** Eσ$^{70}$ cryo-EM processing pipeline.
**Figure supplement 4.** Eσ$^{70}$ cryo-EM.

residue S2) is only 4.3 Å from the active site Mg$^{2+}$ (*Figure 1E*). TraR$_G$ interacts primarily with the β'rim-helices at the entrance of the secondary channel (*Figure 1D*).

The interactions of TraR$_C$ with RNAP differ substantially between the cryo-EM and X-ray structures due to conformational changes induced by TraR binding detected by the cryo-EM structure that were not observed in the X-ray structure (see below). Indeed, the cryo-EM and X-ray structures superimpose with an rmsd of 4.26 Å over 3,471 α-carbons, indicating significant conformational differences.

Cryo-EM data for Eσ$^{70}$ resolved to a nominal resolution of 4.1 Å (*Figure 1—figure supplements 3* and *4*; *Supplementary file 1*). Analysis of the *rpsT* P2-RPo cryo-EM data (*Figure 2*) gave rise to two conformational classes that differed only in the disposition of the upstream promoter DNA and αCTDs (*Figure 2—figure supplement 1*). We focus here on the highest resolution class at a nominal

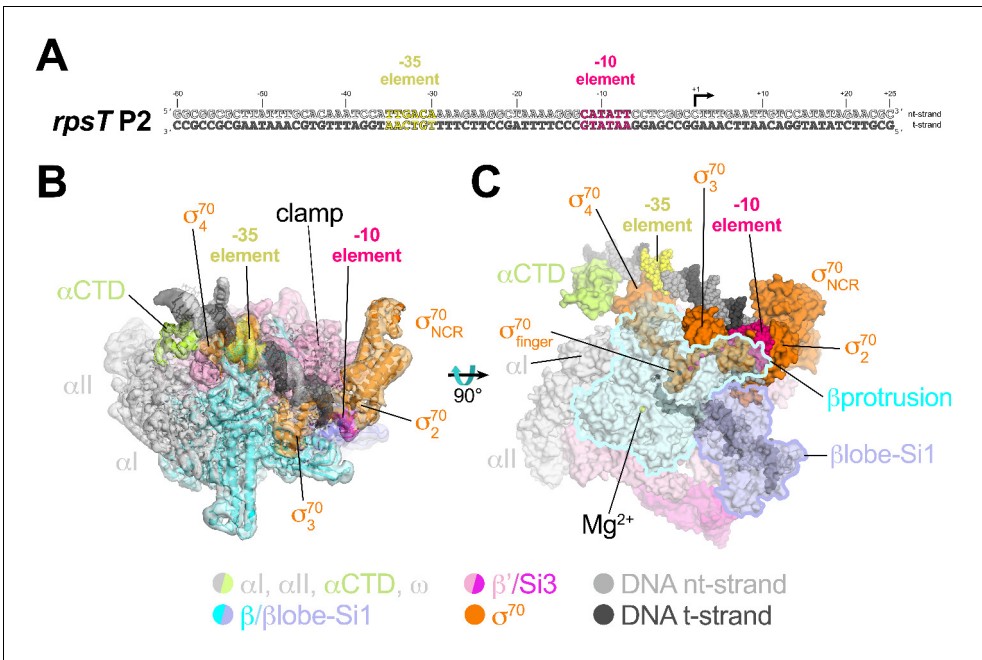

**Figure 2.** Cryo-EM structure of *rpsT* P2-RPo. (**A**) The *Eco rpsT* P2 promoter fragment used for cryo-EM. (**B**) *rpsT* P2-RPo cryo-EM density map (3.4 Å nominal resolution, low-pass filtered to the local resolution) is shown as a transparent surface and colored according to the key. The final model is superimposed. The DNA was modeled from −45 to +21. The t-strand DNA from −10 to −2, and the nt-strand DNA from −3 to +2 were disordered. (**C**) Top view of *rpsT* P2-RPo. DNA is shown as atomic spheres. Proteins are shown as molecular surfaces. Much of the β subunit is transparent to reveal the active site Mg$^{2+}$ (yellow sphere), σ$^{70}_{finger}$, and DNA inside the RNAP active site cleft.

The online version of this article includes the following figure supplement(s) for figure 2:

**Figure supplement 1.** *rpsT* P2-RPo cryo-EM processing pipeline.
**Figure supplement 2.** *rpsT* P2-RPo cryo-EM.

resolution of 3.4 Å (*Figure 2—figure supplement 1*, *Figure 2—figure supplement 2*; *Supplementary file 1*).

The closed-clamp RNAP in the *rpsT* P2-RPo cryo-EM structure interacts with the promoter DNA in the same way as in RPo structures determined by X-ray crystallography (*Bae et al., 2015b*; *Bae et al., 2015a*; *Hubin et al., 2017b*) or cryo-EM (*Boyaci et al., 2019*) and is consistent with the DNase I footprint of the *rpsT* P2 RPo (*Gopalkrishnan et al., 2017*). In the *rpsT* P2-RPo structure we observed an α-subunit C-terminal domain [αCTD; *Ross et al., 1993*] bound to the promoter DNA minor groove (*Benoff et al., 2002*; *Ross et al., 2001*) just upstream of the promoter −35 element [−38 to −43, corresponding to the proximal UP element subsite; *Estrem et al., 1999*]. This αCTD interacts with $\sigma^{70}_4$ through an interface previously characterized through genetic analyses (*Ross et al., 2003*) (*Figure 2B,C*). The αCTDs are linked to the α-N-terminal domains (αNTDs) by ~15 residue flexible linkers (*Blatter et al., 1994*; *Jeon et al., 1995*). Density for the residues connecting the αCTD and αNTD was not observed in the cryo-EM map.

Comparing the RNAP conformations of the TraR-Eσ70, Eσ70, and *rpsT* P2-RPo cryo-EM structures revealed key differences that suggest how TraR activates and inhibits transcription. Below we outline these differences and test their implications for function.

## β′Si3 is in two conformations, one of which is important for TraR activation function

The three TraR-Eσ70 structures differ from each other only in the disposition of Si3. Si3 comprises two tandem repeats of the sandwich-barrel hybrid motif (SBHM) fold (*Chlenov et al., 2005*; *Iyer et al., 2003*), SBHMa and SBHMb (*Figure 3A*). Si3 is linked to the TL-helices by extended, flexible linkers. In TraR-Eσ70(I) and TraR-Eσ70(II), Si3 is in two distinct positions with respect to the RNAP (*Figures 1A, B* and *3A*), while in TraR-Eσ70(III) Si3 is disordered (*Figure 1—figure supplement 1D*). Si3 in the TraR-Eσ70(I) structure [Si3(I)] interacts primarily with the β′shelf (SBHMa) and the β′jaw (SBHMb) in a manner seen in many previous *Eco* RNAP X-ray (*Bae et al., 2013*) and cryo-EM structures (*Chen et al., 2017*; *Kang et al., 2017*; *Liu et al., 2017*). Si3 in the TraR-Eσ70(II) structure [Si3 (II)] is rotated 121° such that SBHMa interacts with the β′jaw and SBHMb interacts with TraR$_G$ (*Figure 3A–C*), a disposition of Si3 that, to our knowledge, has not been observed previously.

To test if this alternative conformation [Si3(II)] is relevant to TraR function, we compared TraR-mediated function with wild-type (WT) and ΔSi3-RNAPs at promoters where TraR inhibits or activates transcription. Deletion of Si3 had little to no effect on TraR-mediated inhibition of *rrnB* P1 and *rpsT* P2 (*Figure 3D*, *Figure 3—figure supplement 1A*) but transcription by ΔSi3-RNAP was activated only ~50% compared with WT-RNAP on three different TraR-activated promoters (p*thrABC*, *Figure 3E*; p*argI*, *Figure 3—figure supplement 1B*; p*hisG*, *Figure 3—figure supplement 1C*).

Three TraR$_G$ residues (TraR-E46, R49, and K50) are central to the Si3-TraR$_G$ interface (*Figure 3B, C*). Individual alanine substitutions of these TraR residues (TraR-E46A, R49A, or K50A) gave rise to similar results as deleting Si3. Inhibition of *rrnB* P1 was similar to WT-TraR for TraR-K50A, and mildly impaired for TraR-E46A or R49A (*Figure 3F*; legend for IC$_{50}$ values). Maximal inhibition was achieved at higher E46A or R49A TraR concentrations. However, these same variants exhibited at least ~2 fold reduced activation at the *thrABC* promoter (*Figure 3G*) even at saturating TraR concentrations, indicating a role for the TraR-Si3 interaction in the mechanism of activation. Consistent with these results, these TraR variants were proficient in RNAP binding in a competition assay (*Figure 3—figure supplement 1F*). By contrast, substitutions for nearby TraR variants P43A and P45A were defective for binding to RNAP, and their functional defects were overcome at higher TraR concentrations (*Figure 3—figure supplement 1D–F*).

The combination of the TraR-Si3 interface and the ΔSi3-RNAP mutants was epistatic; the same ~2 fold reduction in activation was observed as with the Si3-TraR interface mutants or the ΔSi3-RNAP mutant individually (*Figure 3—figure supplement 1G*). These results indicate that the Si3(SBHMb)-TraR$_G$ interaction enabled by the Si3(II) conformation accounts for part of the TraR-mediated effect on activation.

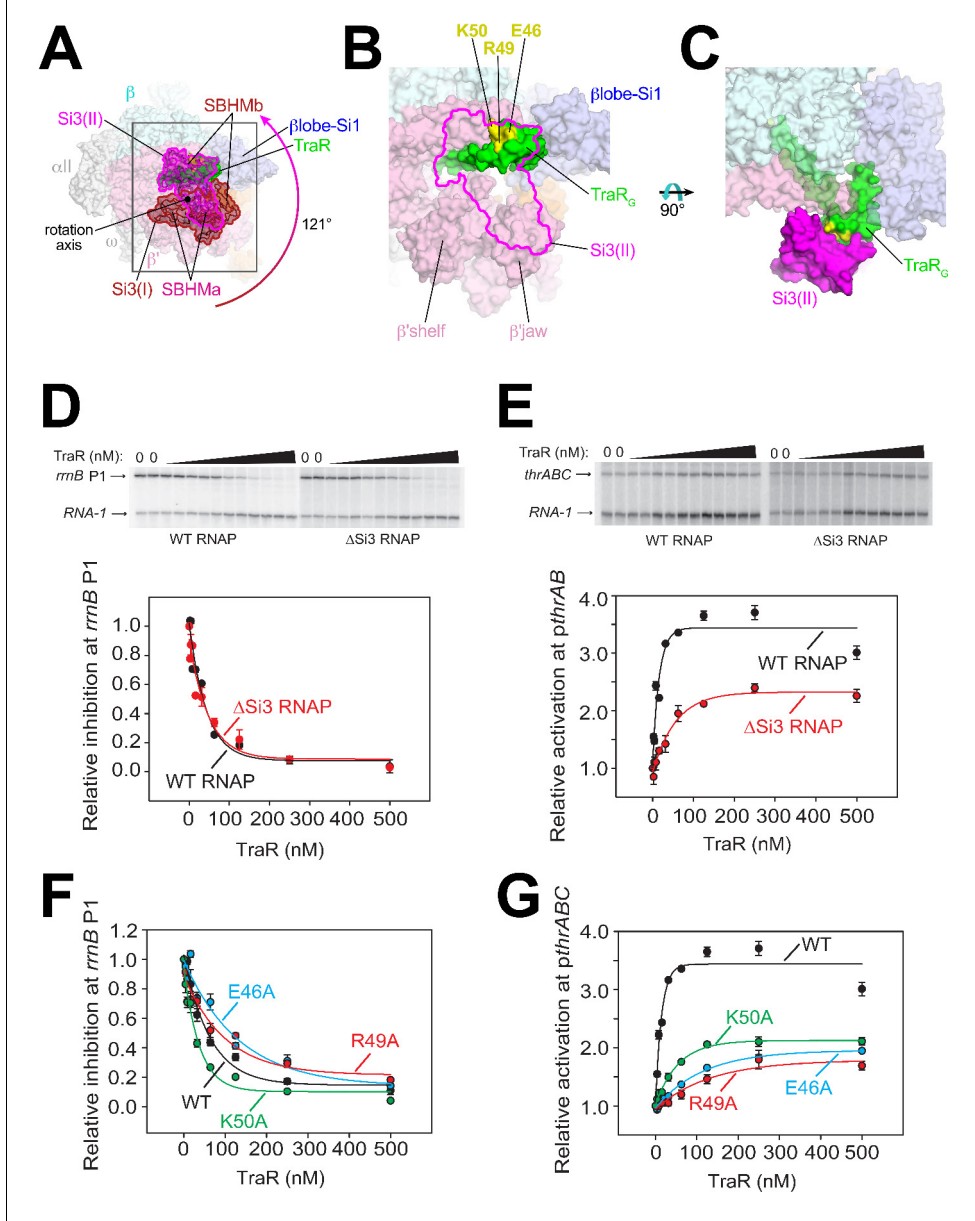

**Figure 3.** Conformational flexibility of β'Si3 in TraR-Eσ70. (**A**) Overall view of TraR-Eσ70 structure with alternative positions of Si3. Si3(I) is shown in brown. A ~ 121° rotation about the rotation axis shown gives rise to the position of Si3(II) shown in magenta. Si3 comprises two SBHM domains (*Chlenov et al., 2005*; *Iyer et al., 2003*), denoted SBHMa and SBHMb. The boxed region is magnified in (**B**). (**B**) Magnified view of TraR-Eσ70(II) [same view as (**A**)]. The position of Si3(II) is outlined in magenta but the rest of Si3 is removed, revealing TraR behind. Three residues central to the TraR-Si3(II) interface (TraR-E46, R49, and K50) are colored yellow. (**C**) Orthogonal view as (**B**), showing the extensive TraR-Si3(II) interface. (**D**) – (**G**) Si3 interaction with TraR$_G$ affects activation but not inhibition. Quantifications show averages with range from two independent experiments. (**D**) (top) Multi round in vitro transcription of *rrnB* P1 over a range of TraR concentrations (wedge indicates 2 nM - 2 μM) in the presence of WT-RNAP or ΔSi3-RNAP as indicated. Plasmid templates also contained the RNA-1 promoter. (bottom) Quantification of transcripts from experiments like those shown on (top) plotted relative to values in the absence of TraR. The IC$_{50}$ for inhibition by TraR was ~40 nM for both data sets. (**E**) (top) Multi round in vitro transcription of *thrABC* over a range of TraR concentrations (wedge indicates 2 nM - 2 μM) in the presence of 20 nM WT-RNAP or ΔSi3-RNAP as indicated. Plasmid templates also contained the RNA-1 promoter. (bottom) Quantification of transcripts from experiments like those shown on (top) plotted relative to values in the absence of TraR. (**F**) and (**G**) Multi round in vitro transcription of *rrnB* P1 (**F**) or p*thrABC* (**G**) was performed with 20 nM WT-Eσ70 at a range of concentrations of WT or variant TraR (2 nM −2 μM). Transcripts were quantified and plotted relative to values in the absence of any

*Figure 3 continued on next page*

*Figure 3 continued*

factor (n = 2). For (**F**) $IC_{50}$ for inhibition by WT-TraR was ~50 nM, by E46A TraR was ~115 nM, R49A TraR was ~85 nM and by K50A TraR was ~30 nM.

The online version of this article includes the following figure supplement(s) for figure 3:

**Figure supplement 1.** RNAP-Si3 interaction with $TraR_G$ residues.

## A TraR-induced ~18° rotation of βlobe-Si1 plays a major role in transcription regulation

The large cleft between the two pincers in the structure of RNAP forms a channel that accommodates the downstream duplex DNA between the β'shelf and the clamp on one side, and the βlobe-Si1 domains on the other (*Figure 2B,C*). In $E\sigma^{70}$ without nucleic acids, this channel is occupied by the $\sigma^{70}_{1.1}$ domain, which is ejected upon entry of the downstream duplex DNA (*Figure 1D*) (*Bae et al., 2013*; *Mekler et al., 2002*). TraR binding induces a ~18° rotation of the RNAP βlobe-Si1 domains (the two domains move together as a rigid body), shifting the βlobe-Si1 towards TraR, allowing the βlobe-Si1 to establish an interface with $TraR_G$ and $TraR_C$ (615 $\mathring{A}^2$ interface area; *Figure 4A*).

Si1 [also called βi4; *Lane and Darst, 2010a*] is an LSI within the βlobe. Most of the TraR/βlobe-Si1 interface (77%) is between TraR and Si1. Deleting Si1 from RNAP nearly abolishes activation function [p*argI*, *Figure 4C*; *thrABC*, *Gopalkrishnan et al., 2017*], even at saturating concentrations of TraR to overcome weakened TraR binding (*Gopalkrishnan et al., 2017*). These results suggest that the βlobe-Si1 rotation induced by TraR is essential to TraR-mediated activation.

The rotation of the βlobe-Si1 widens the gap between the βprotrusion and the βlobe (*Figure 4A*) and changes the shape of the RNAP channel, altering RNAP contacts with $\sigma^{70}_{1.1}$ in $E\sigma^{70}$. We hypothesize that altering the RNAP contacts with $\sigma^{70}_{1.1}$ in the channel facilitates $\sigma^{70}_{1.1}$ ejection during RPo formation, contributing to activation of promoters that are limited at this step. To test this hypothesis, we investigated TraR function on an inhibited (*rrnB* P1) and an activated (*thrABC*) promoter with holoenzyme lacking $\sigma^{70}_{1.1}$ ($E\Delta1.1\sigma^{70}$).

$E\sigma^{70}$ exhibited weak transcription from the *thrABC* promoter in the absence of TraR (referred to here as basal transcription), and transcription from this promoter was stimulated about ~4 fold in the presence of TraR (*Figure 4D*). $E\Delta1.1\sigma^{70}$ exhibited a striking increase in basal transcription activity from this promoter (~32 fold) compared to WT-$E\sigma^{70}$ activity (*Figure 4D*). Only a small further increase in transcription was observed upon the addition of TraR (*Figure 4D*). These results suggest that $\sigma^{70}_{1.1}$ is an obstacle to promoter DNA entering the RNAP channel and that TraR partially overcomes this barrier. In contrast to deletion of region $\sigma^{70}_{1.1}$, which almost entirely bypassed the requirement for TraR, rotation of the βlobe-Si1 did not weaken $\sigma^{70}_{1.1}$-RNAP contacts sufficiently to release $\sigma^{70}_{1.1}$ completely (*Figure 1D*). Rather, we propose that βlobe-Si1 rotation facilitated the competition between promoter DNA and $\sigma^{70}_{1.1}$ during RPo formation. Our results suggest that TraR-activated promoters are defined, in part, by being limited at the $\sigma^{70}_{1.1}$ ejection step, but that the system has evolved to allow activation of WT-RNAP to a level appropriate for the biological need for the gene products, not the maximum level that could be achieved by full ejection of $\sigma^{70}_{1.1}$.

βSi1 was also required for inhibition of *rpsT* P2 (*Figure 4B*) and *rrnB* P1 transcription by TraR (*Gopalkrishnan et al., 2017*). However, in contrast to the effect of $E\Delta1.1\sigma^{70}$ on activated promoters, deletion of $\sigma^{70}_{1.1}$ had little effect on basal transcription from the TraR-inhibited *rrnB* P1 promoter; inhibition of *rrnB* P1 by TraR with $E\Delta1.1\sigma^{70}$ was only slightly defective (*Figure 4—figure supplement 1A,B*). Thus, in contrast to the effects of Si1 on activation by TraR, we suggest that the effect of TraR on inhibition of transcription involves the βlobe-Si1 domains but this is not mediated by $\sigma^{70}_{1.1}$ (see Discussion). We propose that TraR-mediated stimulation of $\sigma^{70}_{1.1}$ release still occurs at inhibited promoters like *rrnB* P1 and *rpsT* P2, but this has little effect on transcription because these promoters are limited by their unstable RPo (*Barker et al., 2001*) (*Figure 4—figure supplement 1*).

In summary, deletion of $\sigma^{70}_{1.1}$ has a major effect on basal transcription of an activated promoter (*Figure 4D*) but only a minor effect on an inhibited promoter (*Figure 4—figure supplement 1A,B*). To qualitatively understand this striking result in terms of kinetic/energetic schemes for transcription initiation from a hypothetical inhibited and activated promoter, we adapted the flux calculator (*Galburt, 2018*) to a four-step linear mechanism (see Materials and methods) that culminates in the

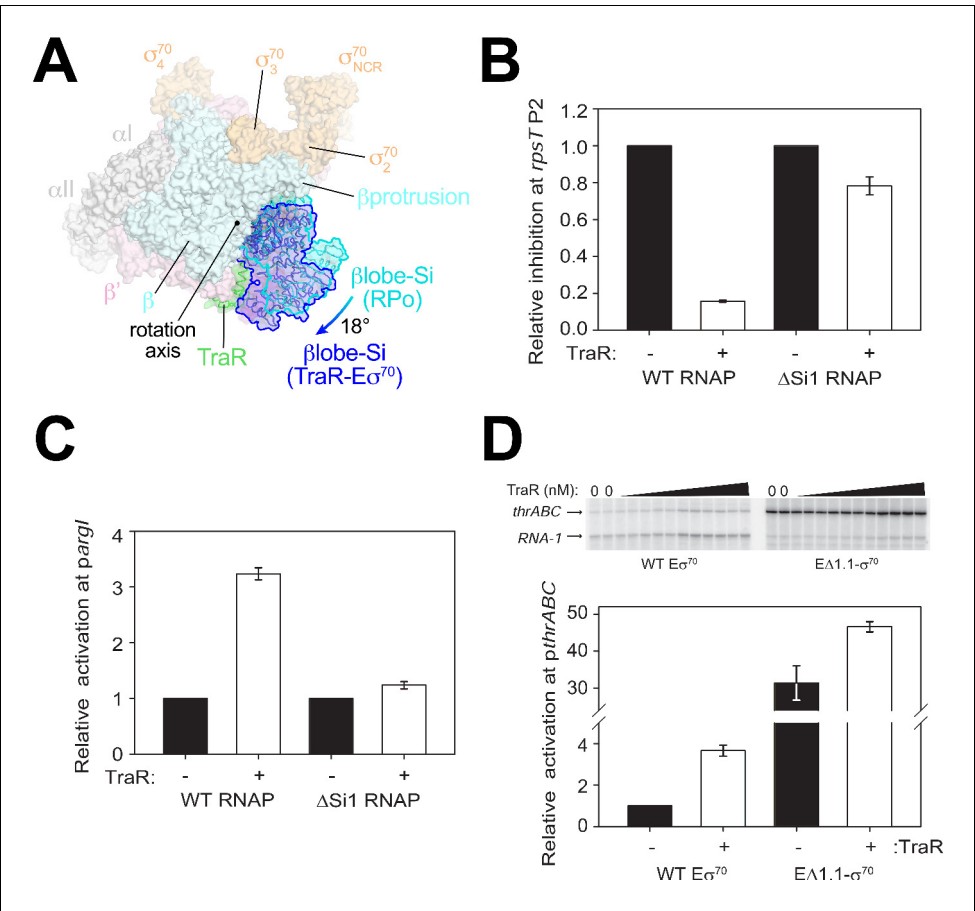

**Figure 4.** TraR and the βlobe-Si1 domain. (**A**) Overall top view of the TraR-Eσ⁷⁰ structure with the βlobe-Si1 in dark blue. The corresponding position of the βlobe-Si1 in the *rpsT* P2-RPo structure (*Figure 2*) is shown in light blue. The βlobe-Si1 of the *rpsT* P2-RPo structure (light blue) undergoes an ~18° rotation about the rotation axis shown to the βlobe-Si1 position in the TraR-Eσ⁷⁰ structure (dark blue), generating an extensive TraR-βlobe-Si1 interface. (**B**) Transcription of inhibited promoter *rpsT* P2 by 20 nM WT-RNAP or ΔSi1-RNAP with (+) or without (-) 250 nM TraR as indicated. Error bars denote standard deviation of three independent measurements. (**C**) Transcription of activated promoter p*argI* by 20 nM WT-RNAP or ΔSi1-RNAP with (+) or without (-) 250 nM TraR as indicated. Error bars denote standard deviation of three independent measurements. (**D**) (top) Multi-round in vitro transcription was carried out at a range of TraR concentrations (wedge indicates 4 nM - 4 μM) in the presence of 20 nM WT-Eσ⁷⁰ or EΔ1.1σ⁷⁰ as indicated. Plasmid template also contained the RNA-1 promoter. (bottom) Transcripts from experiments such as those in (top) were quantified and plotted relative to values in the absence of TraR with WT-Eσ⁷⁰ or EΔ1.1σ⁷⁰ with (+) or without (-) 250 nM TraR as indicated. Averages with range from two independent experiments are shown.

The online version of this article includes the following figure supplement(s) for figure 4:

**Figure supplement 1.** EΔ1.1σ⁷⁰ has small defects for inhibition of *rrnB* P1 by TraR.

---

irreversible generation of RP$_{ITC}$ that has initiated RNA chain synthesis (*Buc and McClure, 1985*; *Roe et al., 1985*; *Sclavi et al., 2005*; *Rutherford et al., 2009*; *Ruff et al., 2015b*; *Hubin et al., 2017a*):

$$R + P \rightleftharpoons RP1 \rightleftharpoons RP2 \rightleftharpoons RPo \rightarrow RP_{ITC}$$

Ejection of σ⁷⁰$_{1.1}$ is thought to be a relatively late step on the pathway to RPo formation (*Ruff et al., 2015a*) so we modeled the effect of deleting σ⁷⁰$_{1.1}$ as lowering the kinetic barrier separating RP2 and RPo (*Figure 4—figure supplement 1C*). The basal energy landscapes for a hypothetical inhibited and activated promoter were modeled after (*Galburt, 2018*) [see Figure 6B

of *Galburt (2018)*. The overall qualitative kinetic/thermodynamic features of inhibited vs. activated promoters are described later in the Discussion. The details of how we used the flux calculator (*Galburt, 2018*) are described in Materials and methods and *Supplementary file 2*. In this scheme, lowering the kinetic barrier separating RP2 and RPo has no effect on transcription output from the negatively regulated promoter but gives rise to a large increase in transcription output from the positively regulated promoter (*Figure 4—figure supplement 1C*), reflecting the experimental results (*Figure 4D*, *Figure 4—figure supplement 1A,B*).

## TraR induces β'shelf rotation and a bridge-helix kink, contributing to inhibition

TraR binding induces a ~ 4.5° rotation of the β'shelf module (*Figure 5A,B*). The BH leads directly into the shelf module, and a kink is introduced in the BH, a long α-helix that traverses the RNAP active site cleft from one pincer to the other, directly across from the active site $Mg^{2+}$ (*Figure 5B,C*). The BH plays critical roles in the RNAP nucleotide addition cycle (*Lane and Darst, 2010b*), including interacting with the t-strand DNA at the active site (*Figure 5D*). TraR causes the BH to kink towards the t-strand DNA (*Figure 5C*), similar to BH kinks observed previously (*Tagami et al., 2011*; *Tagami et al., 2010*; *Weixlbaumer et al., 2013*; *Zhang et al., 1999*), resulting in a steric clash with the normal position of the t-strand nucleotide at +2 (*Figure 5E*). Thus, the TraR-induced BH kink would sterically prevent the proper positioning of the t-strand DNA in RPo, likely contributing to inhibition of transcription.

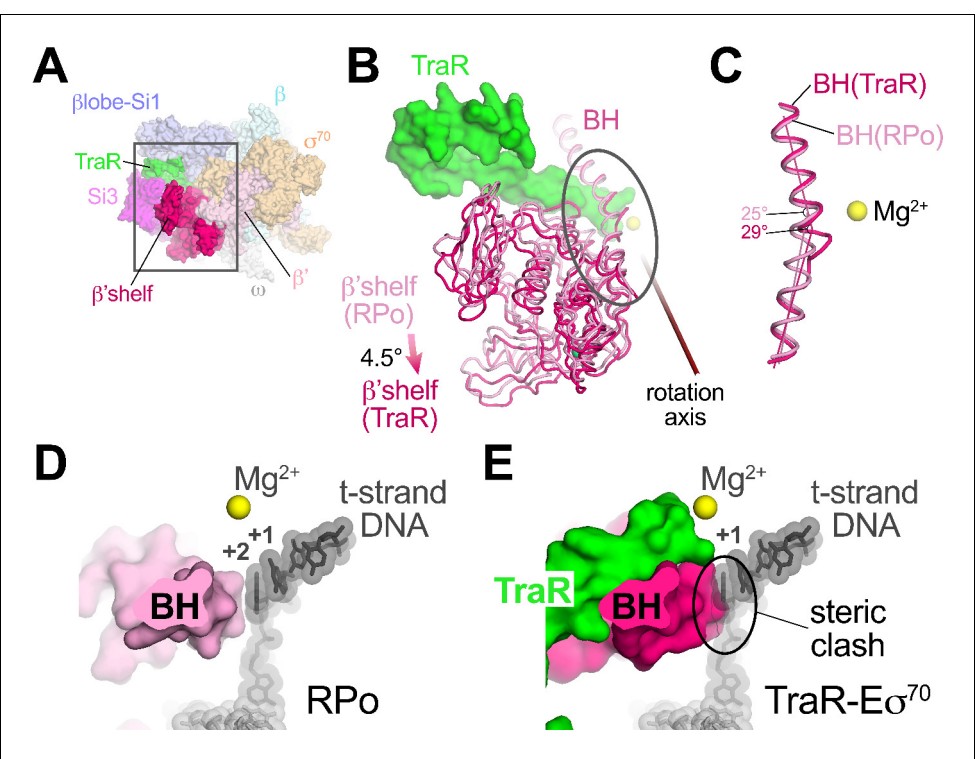

**Figure 5.** TraR rotates the β'shelf and kinks the BH. (**A**) Overall view if the TraR-Eσ[70](I) structure, shown as a molecular surface. The β'shelf domain is highlighted in hot pink. The β'shelf (which here includes the β'jaw) comprises *Eco* β' residues 787-931/1135-1150/1216-1317. The boxed region is magnified in (**B**). (**B**) Comparison of the *rpsT* P2-RPo BH-β'shelf (pink) and the TraR-Eσ[70] BH-β'shelf (hot pink). Binding of TraR induces an ~4.5° rotation (about the rotation axis shown) of the RPo-β'shelf to the position of the TraR-Eσ[70] β'shelf and a kink in the BH (circled region, which is magnified in (**C**)). (**C**) Focus on the region of the BH kink, which is centered near β'L788. The kink in the RPo BH is about 25°, while the kink in the TraR-Eσ[70] BH is about 29°. (**D**) View down the axis of the *rpsT* P2-RPo BH. The t-strand DNA, positioned at the RNAP active site (marked by the $Mg^{2+}$ ion), closely approaches the BH. (**E**) View down the axis of the TraR-Eσ[70] BH. The BH kink induced by TraR binding sterically clashes with the position of the t-strand DNA (superimposed from the RPo structure).

## TraR binding restricts the range of clamp positions in $E\sigma^{70}$

TraR induces conformational changes in the RNAP β'Si3 (*Figure 1A,B*), βlobe-Si1 (*Figure 4A*), β'shelf, and BH (*Figure 5*) structural modules. We noted modest changes in clamp positions (*Supplementary file 3*), but we suspected that conformational heterogeneity of $E\sigma^{70}$ (limiting the resolution of the single particle analysis; *Figure 1—figure supplement 3*) likely arose primarily from a continuous distribution of clamp positions that could not be easily classified into distinct conformational states. We also suspected that the range of clamp positions was dampened in the TraR-$E\sigma^{70}$ and *rpsT* P2-RPo structures. We therefore analysed and compared the heterogeneity of RNAP clamp positions between the $E\sigma^{70}$, TraR-$E\sigma^{70}$, and *rpsT* P2-RPo datasets using multibody refinement as implemented in RELION 3 (*Nakane et al., 2018*). The maps used for multi-body refinement were carefully chosen to be equivalently processed. After initial classification to remove junk particles, particles were 3D auto-refined, then the refinement metadata and post-processing were used as inputs for RELION CTF refinement and Bayesian Polishing (*Zivanov et al., 2018*). After a final round of 3D auto-refinement (but no further classification), the *rpsT* P2-RPo dataset had the smallest number of particles (370,965), so a random subset of particles from the other datasets (TraR-$E\sigma^{70}$ and $E\sigma^{70}$) were processed so that each map for multi-body refinement was generated from the same number of particles (370,965). The final maps used for multi-body refinement had nominal resolutions of 4.0 Å (TraR-$E\sigma^{70}$; red dashed box in *Figure 1—figure supplement 1*), 4.6 Å ($E\sigma^{70}$; red dashed box in *Figure 1—figure supplement 3*), and 3.5 Å (*rpsT* P2-RPo; red dashed box in *Figure 2—figure supplement 1*). We note that dynamic clamp behavior has been observed by single-molecule FRET in solution (*Duchi et al., 2018*).

For $E\sigma^{70}$, three major components (Eigenvectors) of clamp changes were revealed (*Figure 6A–D*). For each Eigenvector, the histogram of Eigenvalues closely approximated a Gaussian distribution (*Figure 6B–D*). To quantitate the range of clamp conformations represented by the Eigenvalues, we divided the particles into three bins according to their Eigenvalues such that each bin contained an equal number of particles (red, gray, and blue in *Figure 6B–D*). Three-dimensional alignments and reconstructions were then calculated for each bin.

For component 1, the red and blue particles gave rise to reconstructions that differed in clamp positions by a rotation angle of 2.7° in a direction we call opening/closing (*Figure 6E*). The low Eigenvalue particles yielded a closed clamp (red), while the high Eigenvalue particles (blue) gave an open clamp. In the middle, the particles having intermediate Eigenvalues (gray) gave a clamp position half-way in between the red and the blue, as expected (not shown).

Component two gave rise to clamp positions that differed by a 4.6° rotation about a rotation axis roughly perpendicular to the open/close rotation axis, a conformational change we call twisting (*Figure 6F*). Finally, component three gave rise to clamp positions that differed by a 2.0° rotation about a third rotation axis parallel with the long axis of the clamp, a conformational change we call rolling (*Figure 6G*).

Using the parameters of the Gaussian fits to the Eigenvalue histograms (*Figure 6B–D*), we could estimate the full range of clamp rotations for each component, which we defined as the rotation range that accounted for 98% of the particles (excluding 1% of the particles at each tail; *Figure 7*).

These same conformational changes (opening/closing, twisting, rolling) were represented in major components of clamp changes for the TraR-$E\sigma^{70}$ and *rpsT* P2-RPo particles as well. The same analyses revealed that TraR binding significantly reduced the range of clamp positions for each of the three clamp motions (*Figure 7*). As expected, the clamp positions for RPo, with nucleic acids stably bound in the downstream duplex channel, were restricted even further for all three of the major clamp motions (*Figure 7*). See Discussion for potential mechanism(s) of effects of clamp conformational changes on transcription.

## Discussion

Our cryo-EM structural analyses show that TraR modulates *Eco* RNAP transcription initiation by binding to and altering the discrete conformation, as well as the conformational heterogeneity, of the RNAP in four major ways: (1) manipulation of the disposition of β'Si3 (*Figures 1A, B* and *3*); (2) alteration of the shape of the RNAP active site cleft through a large rearrangement of the βlobe-Si1 (*Figure 4*); (3) induction of a significant kink in the BH (*Figure 5*); and (4) dampening the range of clamp positions (*Figures 6* and *7*; *Video 1*). A previous crystal structure analysis showed that TraR could

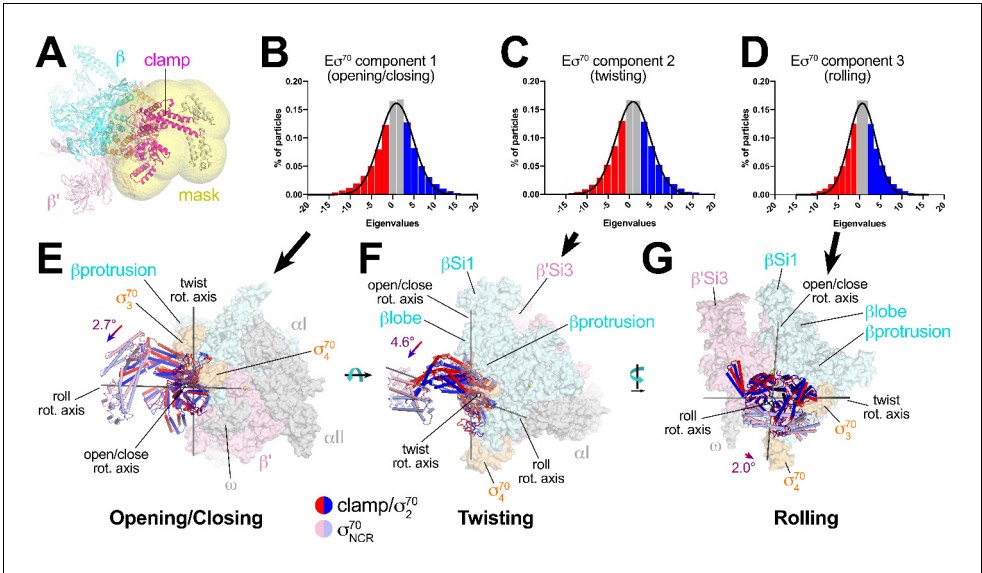

**Figure 6.** Multi-body analysis of Eσ⁷⁰ clamp conformational changes. (**A**) Model of Eσ⁷⁰ refined into the consensus cryo-EM map (nominal 4.1 Å resolution). The RNAP clamp is highlighted in magenta. The clamp (which in the context of Eσ⁷⁰ includes σ⁷⁰₂) comprises the following *Eco* RNAP residues: β 1319–1342; β′ 1–342, 1318–1344; σ⁷⁰92–137, 353–449. The mask used to analyze clamp motions by multi-body refinement (*Nakane et al., 2018*) is shown as a transparent yellow surface. (**B - D**) Histograms of Eigenvalue distributions (% of particles assigned each Eigenvalue from the dataset) for each of the three major principle components (Eigenvectors) from the multi-body analysis (*Nakane et al., 2018*). Each set of particles were divided into three equal-sized bins (red; gray; blue). The solid lines denote Gaussian fits to the histograms. (**B**) Component 1. (**C**) Component 2. (**D**) Component 3. (**E - G**) Three-dimensional reconstructions were calculated from the red and blue-binned particles for each principal component and models were generated by rigid body refinement. The models were superimposed using α-carbons of the RNAP structural core, revealing the alternate clamp positions shown (red and blue α-carbon ribbons with cylindrical helices). The σ⁷⁰ₙCR, attached to the clamp but not included in the clamp motion analyses, is shown in faded colors. For each component, the clamp rotation and the direction of the rotation axis were determined (rotation axes are shown in gray). (**E**) Component 1 - clamp opening/closing. (**F**) Component 2 - clamp twisting. (**G**) Component 3 - clamp rolling.

diffuse into crystals of *Eco* Eσ⁷⁰ and interact with the RNAP β′rim-helices and secondary channel (*Molodtsov et al., 2018*), but none of these four major TraR-mediated conformational changes seen in the cryo-EM analysis presented here were observed in the crystal structure (*Supplementary file 3*). Comparing RNAP conformations, the TraR-Eσ⁷⁰ crystal structure (5W1S) matches the Eσ⁷⁰ crystal structure [4YG2, the same crystal form from which the TraR complex was derived; *Murakami, 2013*] much more closely than the TraR-Eσ⁷⁰ cryo-EM structure (*Supplementary file 3*). Thus, crystal packing constraints prevented the conformation of the RNAP from properly responding to TraR binding.

Our results highlight important advantages of cryo-EM over crystallography for structural analysis of large, conformationally dynamic molecular machines such as RNAP (*Bai et al., 2015a*). First, single-particle cryo-EM analysis does not require crystallization and avoids limitations imposed by crystal packing. Second, multiple, discrete conformational states, such as TraR-Eσ⁷⁰(I), TraR-Eσ⁷⁰(II), and TraR-Eσ⁷⁰(III) (*Figure 1A,B*, *Figure 1—figure supplement 1*), can be revealed from a single sample (*Bai et al., 2015b*). Third, when a conformational change does not parse into discrete states but rather comprises a continuous distribution of conformations, the range of conformational states can nevertheless be assessed experimentally (*Figures 6* and *7*) (*Nakane et al., 2018*).

The consequences of the TraR-induced conformational changes for promoter function (activation or inhibition) depend on the distinctly different properties of the promoters, which are attributable to differences in DNA sequence (*Haugen et al., 2008*; *Sanchez-Vazquez et al., 2019*). Eσ⁷⁰ can complete RPo formation on some promoters in a fraction of a second, whereas RPo's on other promoters require ten minutes or more to form. Likewise, the RPo half-life can vary greatly on different promoters, from a few minutes to many hours. The large range of promoter properties gives rise to

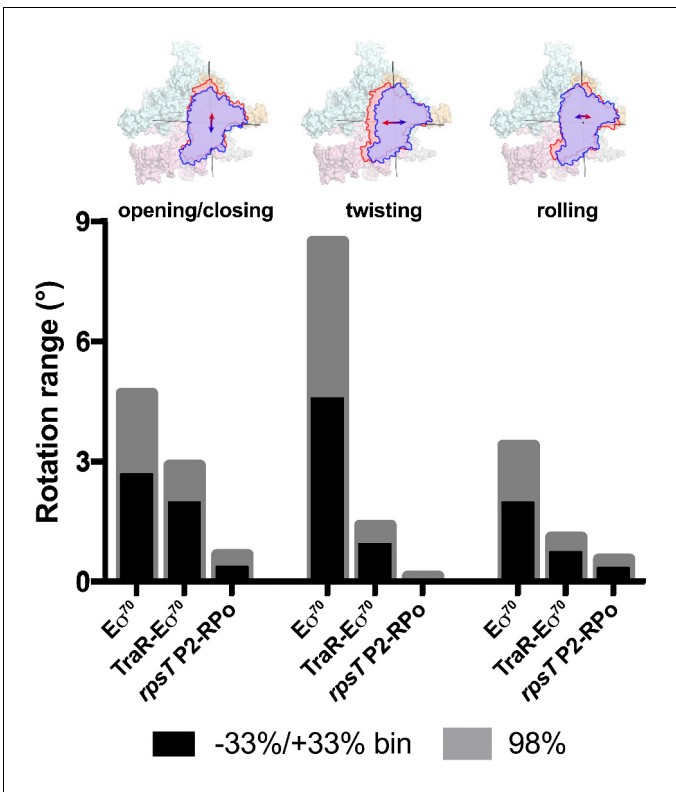

**Figure 7.** Range of clamp conformations for *Eco* RNAP complexes. (top) E$\sigma^{70}$ is shown as a molecular surface ($\alpha$, $\omega$, light gray; $\beta$, light cyan; $\beta'$, light pink; $\sigma^{70}$, light orange) except the clamp/$\sigma^{70}_2$ module is shown schematically as blue or red outlines (the $\sigma^{70}_{NCR}$ is omitted for clarity) to illustrate the direction and approximate range for the three major components of the clamp conformational changes (left, opening/closing; middle, twisting; left, rolling). (bottom) Histograms denote the range of clamp conformational changes for E$\sigma^{70}$, TraR-E$\sigma^{70}$, and *rpsT* P2-RPo, as indicated. The black bars denote the rotation range defined by dividing the Eigenvalue histograms into three equal bins and determining the clamp position for the red and blue bins ($-33$ %/+33% bin; see *Figure 6*). The gray bars denote the estimated rotation range to include 98% of the particles calculated from the Gaussian fits to the Eigenvalue histograms (1% of the particles excluded from each tail; see *Figure 6*).

activities of bacterial transcription initiation that vary over ~4 orders of magnitude and provide rich targets for regulation (*Galburt, 2018*; *McClure, 1985*).

Mechanistic studies of ppGpp/DksA- and TraR-dependent regulation of initiation revealed general characteristics of promoters that are either activated or inhibited by these factors and led to a

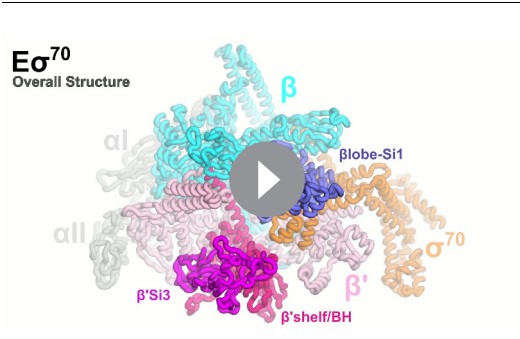

**Video 1.** Video illustrating changes in conformation and conformational dynamics of RNAP induced by TraR binding.
https://elifesciences.org/articles/49375#video1

conceptual model for how they activate some promoters while inhibiting others (*Gopalkrishnan et al., 2017*; *Gourse et al., 2018*; *Paul et al., 2004a*; *Paul et al., 2005*; *Rutherford et al., 2009*). In the absence of factors, inhibited promoters generate RPo very rapidly (*Rao et al., 1994*), but the final transcription-competent RPo is relatively unstable. The half-life of RPo for the inhibited promoter *rrnB* P1 is measured in seconds or a few minutes, depending on solution conditions (*Barker et al., 2001*). In the absence of either transcription factors or high initiating NTP concentrations, RPo at *rrnB* P1 exists in equilibrium with earlier intermediates along the pathway to RPo formation (*Gopalkrishnan et al., 2017*;

*Rutherford et al., 2009*). The very short RPo half-life at inhibited promoters means that initiation of RNA chain synthesis competes with dissociation of RPo. High NTP concentrations can shift the equilibrium in favor of RPo by mass action by populating complexes that follow RPo in the transcription cycle (*Barker and Gourse, 2001*; *Murray et al., 2003*).

By contrast, activated promoters generate RPo very slowly (*Barker et al., 2001*; *Paul et al., 2005*) but the RPo that is ultimately formed is stable. For example, the activated promoters p*argI*, p*hisG*, and p*thrABC* have RPo half-lives measured in many hours [15 hr,>13 hr, and 6.7 hr, respectively (*Barker et al., 2001*).

In order for a transcription factor, such as TraR, to achieve differential regulation (that is, to activate some promoters but inhibit others through the same effects on RNAP), the factor must affect more than one feature of the multi-step pathway of RPo formation (*Galburt, 2018*). In our model for TraR function, TraR acts on all promoters similarly. TraR relieves kinetic barriers to accelerate RPo formation but at the same time stabilizes an intermediate prior to RPo formation (*Galburt, 2018*). Whether TraR activates or inhibits a promoter depends on the basal kinetic landscape for RPo formation at that promoter (*Figure 8*). As summarized below, our structural analysis of the conformational changes imparted on E$\sigma^{70}$ by TraR binding and our biochemical tests of the functional consequences suggested molecular mechanisms for the effect of TraR on the pathway to RPo formation, providing molecular insight into activation and inhibition.

## Structural mechanism for TraR-mediated activation

Together, several lines of structural, biochemical, genetic and kinetic evidence, presented here or published previously, lead to a model for the structural mechanism of activation by TraR. Previous data showed that TraR or its homolog DksA (in conjunction with ppGpp) enhance the transcription output from activated promoters relative to that with RNAP alone (i.e., in the absence of factor; *Paul et al., 2005*; *Gopalkrishnan et al., 2017*). In addition, kinetic analyses demonstrated that the TraR homolog ppGpp/DksA enhances the rate and amount of RPo formation at an activated promoter (*Paul et al., 2005*) primarily by increasing the rates of step(s) subsequent to RNAP binding to the promoter.

Here we show that in the absence of TraR, RNAP with a deletion of $\sigma^{70}_{1.1}$ displayed greatly increased promoter activity on the *thrABC* promoter relative to the activity with WT RNAP (32-fold increase; *Figure 4D*), indicating that the presence of $\sigma^{70}_{1.1}$ in the RNAP channel presents a major barrier to the formation of RPo on activated promoters. This large effect on promoter activity was not observed with *rrnB* P1, an inhibited promoter (*Figure 4—figure supplement 1*). These results are consistent with previous reports that deletion of $\sigma^{70}_{1.1}$ had different effects on different promoters, increasing RPo at some promoters but not at others (*Ruff et al., 2015b*; *Hook-Barnard and Hinton, 2009*; *Vuthoori et al., 2001*).

We suggest that $\sigma^{70}_{1.1}$ poses the most significant barrier to RPo formation at activated promoters, but not the only one, since addition of TraR to RNAP lacking $\sigma^{70}_{1.1}$ resulted in a small TraR-dependent increase in transcription of the *thrABC* promoter (*Figure 4D*). We suggest that this small increase could result from the restriction of clamp motion by TraR (described in the Results section) at a step prior to $\sigma^{70}_{1.1}$ ejection, perhaps transcription bubble nucleation (*Feklistov et al., 2017*).

We propose that TraR binding allosterically alters and weakens $\sigma^{70}_{1.1}$ interactions in the RNAP channel by causing a large (~18°) rotation of the βlobe-Si1 module that forms one wall of the channel (*Figure 4A*; see also *Video 1*). In addition, the interaction of TraR$_G$ with the SBHMa motif of Si3 (*Figure 3*, *Figure 3—figure supplement 1*) may also contribute to weakening of $\sigma^{70}_{1.1}$ interactions with the main channel, because Si3 is inserted within the trigger loop (TL) and its interactions with TraR could perturb interactions of the TL/BH/Switch one region with $\sigma^{70}_{1.1}$. Deletion of β'Si3 has been shown to reduce the lifetime of open complexes by 3–10 fold (*Ruff et al., 2015b*), suggesting the possibility that the reduced activation observed with the Si3 deletion RNAP could also result from effects on RPo stability.

The TraR-induced conformational changes that alter and weaken $\sigma^{70}_{1.1}$ interactions in the main channel are proposed to facilitate displacement of $\sigma^{70}_{1.1}$ by promoter DNA (*Figures 1* and *4*; see also *Video 1*). However, effects of TraR on clamp conformation may also contribute to the increase in RPo by increasing bubble nucleation, as proposed by *Feklistov et al. (2017)*. Together, these effects could be sufficient to account for the TraR-dependent increase in the amount of RPo formed at an activated promoter.

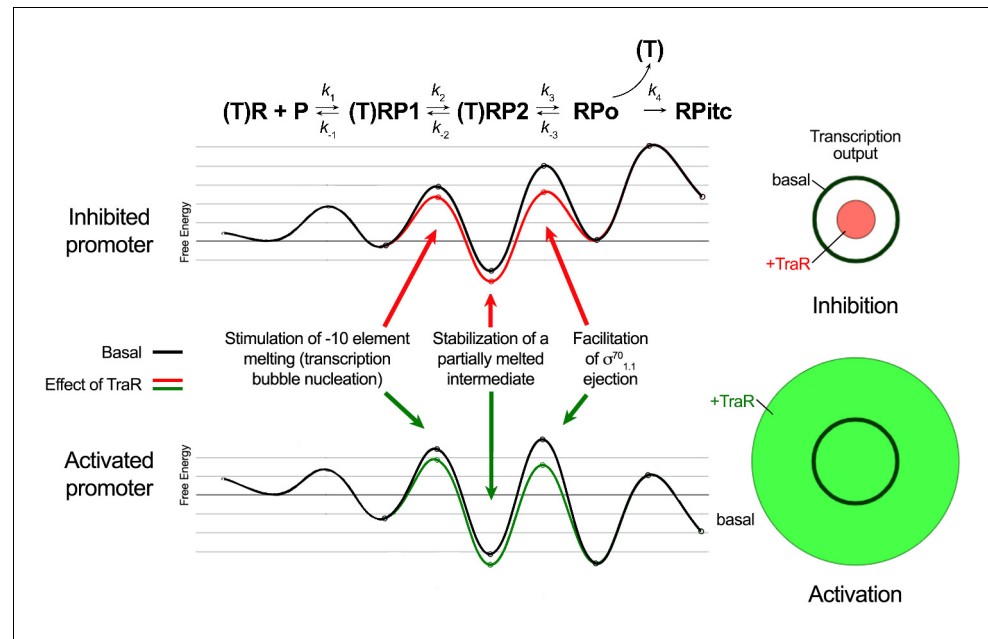

**Figure 8.** Proposed effects of TraR on the free energy diagram for hypothetical inhibited and activated promoters. Shown at the top is a proposed three-step linear kinetic scheme for RPo formation (*Hubin et al., 2017a*) with an added fourth irreversible step (formation of $RP_{itc}$) once RNA synthesis begins. (T) denotes the presence of TraR, which must dissociate to allow the transition of RPo - > RPitc. The basal (WT-RNAP) free energy diagrams for hypothetical inhibited (top) and activated (bottom) promoters are shown in black (adapted from *Galburt, 2018* as described in Materials and methods). The proposed influence of TraR on the energy diagram (lowering the kinetic barrier for the transition RP1 ⇌ RP2; lowering the free energy of RP2 relative to RPo; lowering the kinetic barrier for the transition RP2 ⇌ RPo) is shown (inhibited promoter, red curve; activated promoter, green curve) along with proposed links with the structural effects of TraR binding to RNAP described here. The steady-state transcription output [calculated with the transcription flux calculator (*Galburt, 2018*) is represented by the circles on the right. The precise values for the inputs and outputs for the flux calculator are tabulated in *Supplementary file 2*. The area inside the black circle represents the basal transcription output. The red or green circles (inhibited or activated promoters, respectively) represent the effect of TraR on the transcription output.

Since the structural data indicate that TraR bound to the RNAP complex would sterically block initial NTP access to the active site, along with t-strand positioning and catalysis (*Figure 5*; see also *Molodtsov et al., 2018*), TraR must dissociate from the complex in order to allow initiation from the activated promoter. The proposal that TraR dissociates from the complex at an activated promoter prior to the dissociation of RNAP from the promoter is supported by previous data on the lifetimes of these complexes. The lifetime of RPo at activated promoters (measured in hours; *Barker et al., 2001*) is much longer than the lifetime of the transcription factors on RNAP (GreB, DksA and by extension TraR; measured in seconds; *Stumper et al., 2019*).

The kinetics of transcription initiation are illustrated by a schematic energy landscape in *Figure 8*, where specific structural effects of TraR are correlated with changes in free energy at positively and negatively regulated promoters. The TraR-dependent changes in RNAP conformation and the resulting effects on $\sigma^{70}_{1.1}$ would lower the kinetic barrier to RPo formation. At an activated promoter, the free energy of the transition state between RP2 and RPo is rate-limiting, and it is reduced by the TraR-induced weakening of $\sigma^{70}_{1.1}$ interactions in the main channel, leading to an increase in transcription (*Figure 8*).

## Structural mechanism for TraR-mediated inhibition

TraR (and ppGpp/DksA)-inhibited promoters form RPo rapidly (*Rao et al., 1994*) but their intrinsically unstable RPo results in significant population of earlier intermediates (*Gopalkrishnan et al., 2017*; *Rutherford et al., 2009*). Although TraR likely accelerates bubble nucleation and $\sigma^{70}_{1.1}$ ejection at inhibited promoters, these steps are already rapid and transcription output is not affected. In

contrast to its effects on an activated promoter, deletion of $\sigma^{70}_{1.1}$ had much smaller effects on basal transcription and inhibition by TraR on an inhibited promoter (*Figure 4—figure supplement 1*).

However, TraR binding induces two distinct conformational changes in RNAP that we propose disfavor RPo formation, accounting for inhibition. Most prominent is a direct interaction of TraR with βSi1 that leads to the 18 Å displacement of the βlobe that alters the shape of the main channel and may stabilize DNA contacts in an intermediate prior to RPo (such as RP2; *Figure 8*). Stabilization of an intermediate compared to RPo at inhibited promoters would have a dramatic effect on transcription output by further shifting occupancy by RNAP to earlier intermediates in the RPo formation pathway. Consistent with these hypotheses, ΔSi1-RNAP has reduced TraR-mediated inhibition (*Figure 4B*), and footprints with RNAP on negatively regulated promoters like *rrnB* P1 and *rpsT* P2 have a shortened downstream boundary of DNase I protection (*Gopalkrishnan et al., 2017*).

Inhibition by TraR is multipartite. In addition to causing rotation of the βlobe-Si1, TraR binding also induces a kinked BH that sterically clashes with proper positioning of the t-strand DNA near the active site (*Figure 5*), as reported previously (*Molodtsov et al., 2018*). Precise positioning of the t-strand DNA at the active site is critical for efficient catalysis of phosphodiester bond formation by RNAP in the $S_N2$ mechanism (*Yee et al., 1979*). Occupancy of the secondary channel by TraR would be expected to block catalysis by sterically blocking TL folding required for catalysis and preventing access of initiating NTPs to the active site, ensuring that any complexes containing TraR that proceeded to RPo could not initiate transcription.

Why not attribute inhibition entirely to this ability of TraR to block iNTP access to the active site? Recent single molecule fluorescence studies (*Stumper et al., 2019*) indicated that secondary channel binding factors (GreB, DksA and by extension TraR) formed an RNAP-factor complex prior to binding to promoter DNA, and dissociated together with RNAP from an inhibited promoter complex (*rrnB* P1) because the lifetime of RNAP on *rrnB* P1 is shorter than the lifetime of the factors on RNAP (*Stumper et al., 2019*). Since the affinity of TraR for RNAP is similar to that of DksA (*Gopalkrishnan et al., 2017*), it is likely that TraR exhibits similar properties. Binding of the factors to preformed RPo was not observed (*Stumper et al., 2019*), consistent with the ten-fold reduced affinity of DksA for RPo (*Lennon et al., 2009*). These data are consistent with our model for inhibition, suggesting that TraR would bind together with RNAP to an inhibited promoter and severely reduce RPo formation by stabilizing an intermediate (*Figure 8*). TraR would remain associated with the complex until RNAP dissociated. TraR's presence in the complex for the entire time of RNAP occupancy of the promoter would leave little opportunity for RPo to form and would also prevent catalysis by any RPo that did form.

## TraR manipulates *Eco* RNAP lineage-specific insertions to modulate transcription initiation

The β and β' subunits of the bacterial RNAP are conserved throughout evolution, containing 16 and 11 sequence regions, respectively, common to all bacterial RNAPs (*Lane and Darst, 2010a*). These shared sequence regions are separated by relatively nonconserved spacer regions in which large LSIs can occur (*Lane and Darst, 2010a*). The LSIs are typically independently-folded, highly mobile domains, ranging in size from 50 to 500 amino acids on the RNAP surface. A key feature of the mechanism of TraR function is modulation of *Eco* RNAP transcription initiation through conformational changes brought about by interactions with two of the *Eco* RNAP LSIs, βSi1 (*Figure 4A*) and β'Si3 (*Figure 3A–C*).

*Eco* βSi1 was originally designated dispensable region 1 (*Severinov et al., 1994*), but its deletion reduced growth at 30 °C and prevented growth at 42°, suggesting that it might serve as a binding determinant for regulatory factors (*Artsimovitch et al., 2003*). Indeed, TraR interacts with βSi1 as well as with the nearby βlobe to distort the RNAP channel (*Figure 4A*), effecting both inhibition (*Figure 4B*) and activation (*Figure 4C*) by TraR.

*Eco* β'Si3 is an unusual LSI as it is inserted in the middle of the TL, a key structural element in the RNAP nucleotide addition cycle that is conserved in all multi-subunit RNAPs (*Lane and Darst, 2010a*). As a consequence, Si3 plays a central role in *Eco* RNAP function, and cells deleted for Si3 are not viable (*Artsimovitch et al., 2003*; *Zakharova et al., 1998*). Si3 is highly mobile, rotating about 33° to accommodate folding and unfolding of the TL at each RNAP nucleotide addition cycle (*Malinen et al., 2012*; *Zuo and Steitz, 2015*). Si3 was often disordered in *Eco* RNAP crystal structures [for example, see *Molodtsov et al., 2018*]. In our cryo-EM analysis, TraR engages with Si3,

stabilizing a previously undetected conformation of Si3 affecting activation (*Figure 3*). Si3 has been implicated previously in RPo formation since the Δβ'Si3-RNAP forms an unstable RPo (*Artsimovitch et al., 2003*).

## Conclusion

TraR-like proteins are widespread in proteobacteria, related bacteriophage, and plasmids (*Gopalkrishnan et al., 2017*; *Gourse et al., 2018*). While TraR function in vivo is incompletely understood, TraR engages with RNAP in much the same way as ppGpp/DksA and regulates transcription similarly. The structural and functional analyses described here identify the RNAP conformational changes responsible for the effects of TraR on transcription, deconvoluting the complicated, multifaceted mechanism that distinguishes activation from inhibition. The complex interplay between TraR binding and RNAP conformation and conformational heterogeneity allows TraR to modulate multiple features of the energy landscape of RPo formation (*Figure 8*), which is key to allowing TraR to effect differential regulation across promoter space without directly interacting with DNA. The very similar effects of ppGpp/DksA and TraR on RNAP function (*Gopalkrishnan et al., 2017*; *Gourse et al., 2018*) suggests that DksA-ppGpp works via a similar structural mechanism.

# Materials and methods

**Key resources table**

| Reagent type (species) or resource | Designation | Source or reference | Ident-ifiers | Additional information |
|---|---|---|---|---|
| Strain, strain background (*Escherichia coli*) | *Eco* BL21(DE3) | EMD Millipore (Burlington, MA) | | |
| Recombinant DNA reagent | pACYCDuet-1_Ec_*rpoZ* | PMID: 21416542 | | |
| Recombinant DNA reagent | pEc*rpoABC*(-XH)Z | PMID: 21416542 | | |
| Recombinant DNA reagent | pET28a | EMD Millipore | | |
| Recombinant DNA reagent | pET28a-His$_{10}$-SUMO *rpoD* | PMID: 28988932 | | |
| Recombinant DNA reagent | pET28a-His$_{10}$-SUMO *traR* (pRLG15142) | This paper | | Encodes *Eco* TraR with N-terminal His$_{10}$-SUMO tag (Darst lab) |
| Recombinant DNA reagent | pRLG770 | PMID: 2209559 | | In vitro transcription vector, AmpR |
| Recombinant DNA reagent | pRLG770-*rrnB* P1 (pRLG13065) | PMID: 27237053 | | *rrnB* P1 with −88 to +50 endpoints |
| Recombinant DNA reagent | pRLG770-*argI* (pRLG13098) | PMID: 11162084 | | p*argI* with −45 to +32 endpoints |
| Recombinant DNA reagent | pRLG770-*hisG* (pRLG13099) | PMID: 15899978 | | p*hisG* with −60 to +1 endpoints |
| Recombinant DNA reagent | pRLG770-*rpsT* P2 (pRLG14658) | PMID: 21402902 | | *rpsT* P2 with −89 to +50 endpoints |
| Recombinant DNA reagent | pRLG770-*thrABC* (pRLG15276) | PMID: 11162084 | | p*thrABC* with −72 to +16 endpoints |
| Recombinant DNA reagent | pT7 αββ'(Δ943–1130) (pIA331) | PMID: 12511572 | | ΔSi3 RNAP |
| Recombinant DNA reagent | pIA900 *rpoB* Δ225–343ΩGG (pRLG12586) | PMID: 28652326 | | ΔSi1 RNAP |
| Recombinant DNA reagent | pET28a-His$_{10}$-SUMO P43A *traR* (pRLG14844) | This paper | | Encodes *Eco* TraR[P43A] with N-terminal His$_{10}$-SUMO tag (Gourse lab) |

*Continued on next page*

*Continued*

| Reagent type (species) or resource | Designation | Source or reference | Ident-ifiers | Additional information |
|---|---|---|---|---|
| Recombinant DNA reagent | pET28a- His$_{10}$- SUMO P45A *traR* (pRLG14846) | This paper | | Encodes *Eco* TraR[P45A] with N-terminal His$_{10}$-SUMO tag (Gourse lab) |
| Recombinant DNA reagent | pET28a-His$_{10-}$ SUMO E46A *traR* (pRLG14847) | This paper | | Encodes *Eco* TraR[E46A] with N-terminal His$_{10}$-SUMO tag (Gourse lab) |
| Recombinant DNA reagent | pET28a-His$_{10}$-SUMO R49A *traR* (pRLG15278) | This paper | | Encodes *Eco* TraR[R49A] with N-terminal His$_{10}$-SUMO tag (Gourse lab) |
| Recombinant DNA reagent | pET28a-His$_{10}$-SUMO K50A *traR* (pRLG15279) | This paper | | Encodes *Eco* TraR[K50A] with N-terminal His$_{10}$-SUMO tag (Gourse lab) |
| Sequence-based reagent | P43A *traR* | IDT, this paper | | 5' GAAGCATGCGGAAATGCTATTCCGGAAGCC 3' (Gourse lab) |
| Sequence-based reagent | P45A *traR* | IDT, this paper | | 5' GGAAATCCTATTGCGGAAGCCCGGCGG 3' (Gourse lab) |
| Sequence-based reagent | E46A *traR* | IDT, this paper | | 5' GGAAATCCTATTCCGGCAGCCCGGCGGAAAATA 3' (Gourse lab) |
| Sequence-based reagent | R49A *traR* | IDT, this paper | | 5' ATTCCGGAAGCCCGGGCGAAAATATTTCCCGGT 3' (Gourse lab) |
| Sequence-based reagent | K50A *traR* | IDT, this paper | | 5' ATTCCGGAAGCCCGGCGGGCAATATTTCCCGGT 3' (Gourse lab) |
| Sequence-based reagent | SumoF | IDT, this paper | | 5' GGGGAATTGTGAGCGGATAACAATTCC 3' (Gourse lab) |
| Sequence-based reagent | SumoR | IDT, this paper | | 5' GTCCCATTCGCCAATCCGGATATAG 3' (Gourse lab) |
| Sequence-based reagent | TraR_sumo_vector_FOR | IDT, this paper | | 5'-AAACATTATGCATAACAAAGCCCGAAAGGAAGCTGAG—3' (Gourse lab) |
| Sequence-based reagent | pETsumo_traR_vector_REV | IDT, this paper | | 5'-CGGCTTCATCACTTCCACCAATCTGTTCTCTGTGAGCC—3' (Gourse lab) |
| Sequence-based reagent | TraR_sumo_fragment_REV | IDT, this paper | | 5'-TCGGGCTTTGTTATGCATAATGTTTTCTCTGTCTTTCCTGATACG—3' (Gourse lab) |
| Sequence-based reagent | TraR_sumo_fragment_FOR | IDT, this paper | | 5'-CAGATTGGTGGAAGTGATGAAGCCGATGAAGCATATTCAG—3' (Gourse lab) |
| Sequence-based reagent | *rrnB*P1(−63 to +20) top | IDT, this paper | | 5'-GGTCAGAAAATTATTTTAAATTTCCTCTTGTCAGGCCGGAATAACTCCCTATAATGCGCCACCACTGACACGGAACAACGGCG—3' (Darst lab) |
| Sequence-based reagent | *rrnB*P1(−63 to +20) bot | IDT, this paper | | 5'-CGCCGTTGTTCCGTGTCAGTGGTGGCGCATTATAGGGAGTTATTCCGGCCTGACAAGAGGAAATTTAAAATAATTTTCTGACC—3' (Darst lab) |

*Continued on next page*

*Continued*

| Reagent type (species) or resource | Designation | Source or reference | Ident-ifiers | Additional information |
|---|---|---|---|---|
| Sequence-based reagent | *rps*TP2(−60 to +25) top | IDT, this paper | | 5'-GGCGGCGCTTATTTGCACAAATCCATTGACAAAAGAAGGCTAAAAGGGCATATTCCTCGGCCTTTGAATTGTCCATATAGAACGC −3' (Darst lab) |
| Sequence-based reagent | *rps*TP2 (−60 to +25) bot | IDT, this paper | | 5'-GCGTTCTATATGGACAATTCAAAGGCCGAGGAA TAT GCCCTTTTAGCCTTCTTTTGTCAATGGATTTGT GCAAATAAGCGCCGCC −3' (Darst lab) |
| Chemical compound, drug | 3-[(3-Cholamidopropyl) dimethylammonio]−2-Hydroxy-1-Propanesulfonate (CHAPSO) | Anatrace | Cat# C317 | |
| Software, algorithm | Bayesian Polishing | PMID: 30412051 | | |
| Software, algorithm | Bsoft | PMID: 23954653 | | |
| Software, algorithm | Coot | PMID: 15572765 | | |
| Software, algorithm | cryoSPARC | PMID: 28165473 | | |
| Software, algorithm | CTFFIND4 | PMID: 26278980 | | |
| Software, algorithm | EMAN2 | PMID: 16859925 | | |
| Software, algorithm | Gautomatch | http://www.mrc-lmb.cam.ac.uk/kzhang/Gautomatch | | |
| Software, algorithm | Gctf | PMID: 26592709 | | |
| Software, algorithm | Molprobity | PMID: 20057044 | | |
| Software, algorithm | MotionCor2 | PMID: 28250466 | | |
| Software, algorithm | Multi-body refinement | PMID: 29856314 | | |
| Software, algorithm | PHENIX | PMID: 20124702 | | |
| Software, algorithm | RELION | PMID: 23000701 | | |
| Software, algorithm | SerialEM | PMID: 16182563 | | |
| Software, algorithm | UCSF Chimera | PMID: 15264254 | | |
| Software, algorithm | Unblur | PMID: 26023829 | | |
| Other | C-flat CF-1.2/1.3 400 mesh gold grids | Electron Microscopy Sciences | Cat# CF413-100-Au | |

## Strains, Plasmids and Primer sequences

Plasmids are listed in *Supplementary file 4* and oligonucleotide and geneblock sequences are in *Supplementary file 5*. Bacteria were grown in LB Lennox media or on LB agar plates. Media was supplemented with ampicillin (100 μg/ml) or kanamycin (30 μg/ml) if needed. TraR was made by cloning the *traR* gene in a pET28-based His$_{10}$-SUMO vector which allowed removal of the cleavable

N-terminal $His_{10}$-SUMO tag with Ulp1 protease. ESI-Mass Spectrometry revealed that the molecular mass of purified TraR corresponded to that of a monomer lacking the N-terminal methionine [Figure S6 of *Gopalkrishnan et al., 2017*], hence *traR* without the initial M was cloned into the SUMO vector. This tag-less version of TraR exhibited the same level of activity as a previous TraR construct with four additional residues (LVPR) at the C-terminal end leftover after $His_6$ tag cleavage in the TraR-thrombin site-$His_6$ construct (*Gopalkrishnan et al., 2017*).

## Expression and purification of TraR for cryo-EM

The $His_{10}$-SUMO-TraR plasmid was transformed into competent *Eco* BL21(DE3) by heat shock. The cells were grown in the presence of 25 µg/mL kanamycin to an $OD_{600}$ of 0.5 in a 37°C shaker. TraR expression was induced with a final concentration of 1 mM isopropyl ß-D-thiogalactopyranoside (IPTG) for 3 hr at 37°C. Cells were harvested by centrifugation and resuspended in 50 mM Tris-HCl, pH 8.0, 250 mM NaCl, 5 mM imidazole, 10% glycerol (v/v), 2.5 mM dithiothreitol (DTT), 10 µM $ZnCl_2$, 1 mM phenylmethylsulfonyl fluoride (PMSF, Sigma-Aldrich, St. Louis, MO), 1x protease inhibitor cocktail (PIC, Sigma-Aldrich). Cells were homogenized using a continuous-flow French Press (Avestin, Ottawa, ON, Canada) at 4°C and the resulting lysate was centrifuged to isolate the soluble fraction. The supernatant was loaded onto two 5 mL HiTrap IMAC HP columns (GE Healthcare, Pittsburgh, PA) for a total column volume (CV) of 10 mL. $His_{10}$-SUMO-TraR was eluted at 300 mM imidazole in Ni-column buffer [50 mM Tris-HCl, pH 8.0, 500 mM NaCl, 10% glycerol (v/v), 10 µM $ZnCl_2$, 2.5 mM DTT]. Peak fractions were combined, treated with ULP1 SUMO-protease overnight, and dialyzed against 20 mM Tris-HCl, pH 8.0, 5% glycerol (v/v), 0.1 mM ethylenediaminetetraacetic acid (EDTA), 500 mM NaCl, 10 µM $ZnCl_2$, 2.5 mM DTT, resulting in a final imidazole concentration of 25 mM. The ULP1-cleaved sample was loaded onto one 5 mL HiTrap IMAC HP column to remove $His_{10}$-SUMO-tag along with any remaining uncut TraR. Tagless TraR was collected in the flowthrough and concentrated by centrifugal filtration (Amicon Ultra, EMD Millipore, Burlington, MA). The sample was purified in a final step on a HiLoad 16/60 Superdex 200 column (GE Healthcare). Purified TraR was concentrated to 16 mg/mL by centrifugal filtration, flash-frozen in liquid $N_2$, and stored at −80°C.

## *Eco* $His_{10}$-PPX-RNAP expression and purification

A pET-based plasmid overexpressing each subunit of RNAP (full-length α, β, ω) as well as β'-PPX-$His_{10}$ (PPX; PreScission protease site, LEVLFQGP, GE Healthcare) was co-transformed with a pACYC-Duet-1 plasmid containing *Eco* rpoZ into *Eco* BL21(DE3). The cells were grown in the presence of 100 µg/mL ampicillin and 34 µg/mL chloramphenicol to an $OD_{600}$ of 0.6 in a 37°C shaker. Protein expression was induced with 1 mM IPTG (final concentration) for 4 hr at 30°C. Cells were harvested by centrifugation and resuspended in 50 mM Tris-HCl, pH 8.0, 5% glycerol (v/v), 10 mM DTT, 1 mM PMSF, and 1x PIC. After French Press lysis at 4°C, the lysate was centrifuged twice for 30 min each. Polyethyleneimine [PEI, 10% (w/v), pH 8.0, Acros Organics - ThermFisher Scientific, Waltham, MA] was slowly added to the supernatant to a final concentration of ~0.6% PEI with continuous stirring. The mixture was stirred at 4°C for an addition 25 min, then centrifuged for 1.5 hr at 4°C. The pellets were washed three times with 50 mM Tris-HCl, pH 8.0, 500 mM NaCl, 10 mM DTT, 5% glycerol (v/v), 1 mM PMSF, 1x PIC. For each wash, the pellets were homogenized then centrifuged again. RNAP was eluted by washing the pellets three times with 50 mM Tris-HCl, pH 8.0, 1 M NaCl, 10 mM DTT, 5% glycerol (v/v), 1x PIC, 1 mM PMSF. The PEI elutions were combined and precipitated with ammonium sulfate overnight. The mixture was centrifuged and the pellets were resuspended in 20 mM Tris-HCl, pH 8.0, 1 M NaCl, 5% glycerol (v/v), 5 mM DTT. The mixture was loaded onto three 5 mL HiTrap IMAC HP columns for a total CV of 15 ml. RNAP(β'-PPX-$His_{10}$) was eluted at 250 mM imidazole in Ni-column buffer. The eluted RNAP fractions were combined and dialyzed against 20 mM Tris-HCl, pH 8.0, 100 mM NaCl, 5% glycerol (v/v), 5 mM DTT. The sample was then loaded onto a 35 mL Biorex-70 column (Bio-Rad, Hercules, CA), washed with 10 mM Tris-HCl, pH 8.0, 0.1 mM EDTA, 5% glycerol (v/v), 5 mM DTT] in a gradient from 0.2 M to 0.7 M NaCl. The eluted fractions were combined, concentrated by centrifugal filtration, then loaded onto a 320 mL HiLoad 26/600 Superdex 200 column (GE Healthcare) equilibrated in gel filtration buffer [10 mM Tris-HCl, pH 8.0, 0.1 mM EDTA, 0.5 M NaCl, 5% glycerol (v/v), 5 mM DTT]. The eluted RNAP was supplemented with glycerol to 20% (v/v), flash frozen in liquid $N_2$, and stored at −80°C.

### *Eco* His$_{10}$-SUMO-$\sigma^{70}$ expression and purification

Plasmid encoding *Eco* His$_{10}$-SUMO-$\sigma^{70}$ was transformed into *Eco* BL21(DE3) by heat shock. The cells were grown in the presence of 50 µg/mL kanamycin to an OD$_{600}$ of 0.6 at 37°C. Protein expression was induced with 1 mM IPTG for 1 hr at 30°C. Cells were harvested by centrifugation and resuspended in 20 mM Tris-HCl, pH 8.0, 5% glycerol (v/v), 500 mM NaCl, 0.1 mM EDTA, 5 mM imidazole, 0.5 mM 2-mercaptoethanol (BME), 1 mM PMSF, 1x PIC. After French Press lysis at 4°C, cell debris was removed by centrifugation. The lysate was loaded onto two 5 mL HiTrap IMAC HP for a total CV of 10 ml. His$_{10}$-SUMO-$\sigma^{70}$ was eluted at 250 mM imidazole in 20 mM Tris-HCl, pH 8.0, 500 mM NaCl, 0.1 mM EDTA, 5% glycerol (v/v), 0.5 mM BME. Peak fractions were combined, cleaved with ULP1, and dialyzed against 20 mM Tris-HCl pH 8.0, 500 mM NaCl, 0.1 mM EDTA, 5% glycerol (v/v), 0.5 mM BME, resulting in a final imidazole concentration of 25 mM. The cleaved sample was loaded onto one 5 mL HiTrap IMAC HP to remove His$_{10}$-SUMO-tag along with any remaining uncut $\sigma^{70}$. Tagless $\sigma^{70}$ was collected in the flowthrough and concentrated by centrifugal filtration. The sample was then loaded onto a HiLoad 16/60 Superdex 200 in gel filtration buffer. Peak fractions of $\sigma^{70}$ were pooled, supplemented with glycerol to a final concentration of 20% (v/v), flash-frozen in liquid N$_2$, and stored at −80°C.

### Preparation of E$\sigma^{70}$ for cryo-EM

E$\sigma^{70}$ was formed by mixing purified RNAP and 2.5-fold molar excess of $\sigma^{70}$ and incubating for 20 min at 37°C. E$\sigma^{70}$ was purified on a Superose 6 Increase 10/300 GL column (GE Healthcare) in gel filtration buffer (10 mM Tris-HCl, pH 8.0, 200 mM KCl, 5 mM MgCl$_2$, 10 µM ZnCl$_2$, 2.5 mM DTT). The eluted E$\sigma^{70}$ was concentrated to ~10 mg/mL (~21 µM) by centrifugal filtration (Amicon Ultra).

### Preparation of TraR-E$\sigma^{70}$ for cryo-EM

E$\sigma^{70}$ was formed by mixing purified RNAP and a 2-fold molar excess of $\sigma^{70}$ and incubating for 15 min at room temperature. E$\sigma^{70}$ was purified over a Superose 6 Increase 10/300 GL column in gel filtration buffer. The eluted E$\sigma^{70}$ was concentrated to ~5.0 mg/mL (~10 µM) by centrifugal filtration. Purified TraR was added (5-fold molar excess over RNAP) and the sample was incubated for 15 min at room temperature. An *rrnB* P1 promoter fragment (Integrated DNA Technologies, Coralville, IA) was added (2-fold molar excess over RNAP) and the sample was incubated for a further 15 min at room temperature. The *rrnB* P1 promoter fragment did not bind to TraR-E$\sigma^{70}$ under the cryo-EM grid preparation conditions - the subsequent structural analyses did not reveal any evidence of promoter binding.

### Preparation of *rpsT* P2-RPo for cryo-EM

E$\sigma^{70}$ was prepared as described for TraR-E$\sigma^{70}$, but after the size exclusion chromatography the complex was concentrated to ~10 mg/mL (~20 µM) by centrifugal filtration. Duplex *rpsT* P2 promoter fragment (−60 to +25, *Figure 2A*, IDT) was added to the concentrated E$\sigma^{70}$ to 3-fold molar excess. The sample was incubated for 20 mins at room temperature prior to cryo-EM grid preparation.

### Cryo-EM grid preparation

CHAPSO {3-([3-cholamidopropyl]dimethylammonio)−2-hydroxy-1-propanesulfonate} (Anatrace, Maumee, OH) was added to the samples to a final concentration of 8 mM (*Chen et al., 2019*). The final buffer condition for all the cryo-EM samples was 10 mM Tris-HCl, pH 8.0, 100 mM KCl, 5 mM MgCl$_2$, 10 µM ZnCl$_2$, 2.5 mM DTT, 8 mM CHAPSO. C-flat holey carbon grids (CF-1.2/1.3-4Au) were glow-discharged for 20 s prior to the application of 3.5 µL of the samples. Using a Vitrobot Mark IV (FEI, Hillsboro, OR), grids were blotted and plunge-froze into liquid ethane with 100% chamber humidity at 22°C.

### Acquisition and processing of TraR-E$\sigma^{70}$ cryo-EM dataset

Grids were imaged using a 300 keV Krios (FEI) equipped with a K2 Summit direct electron detector (Gatan, Pleasanton, CA). Datasets were recorded with Serial EM (*Mastronarde, 2005*) with a pixel size of 1.3 Å over a defocus range of 0.8 µm to 2.4 µm. Movies were recorded in counting mode at eight electrons/physical pixel/second in dose-fractionation mode with subframes of 0.3 s over a 15 s exposure (50 frames) to give a total dose of 120 electrons/physical pixel. Dose-fractionated movies

were gain-normalized, drifted-corrected, summed, and dose-weighted using MotionCor2 (*Grant and Grigorieff, 2015*; *Zheng et al., 2017*). CTFFIND4 (*Rohou and Grigorieff, 2015*) was used for contrast transfer function estimation. Particles were picked using Gautomatch (http://www.mrc-lmb.cam.ac.uk/kzhang/) using a 2D template. Picked particles were extracted from the dose-weighted images with RELION (*Zivanov et al., 2018*) using a box size of 256 pixels. Two TraR-Eσ$^{70}$ datasets were collected: dataset 1 consisted of 1546 motion-corrected images with 631,880 particles and dataset 2 consisted of 2132 motion-corrected images with 378,987 particles. The particles from each dataset were curated using RELION 3D classification (N = 3) using a cryoSPARC ab-initio reconstruction (*Punjani et al., 2017*) generated from a subset of the particles. The highest resolution classes from each dataset were subjected to RELION 3D auto-refinement resulting in a 4.69 Å resolution map from dataset 1 and a 4.38 Å resolution map from dataset 2. Refinement metadata and post-processing were used as inputs for RELION CTF refinement and Bayesian Polishing (*Zivanov et al., 2018*). The polished particles from both datasets were combined, resulting in 372,670 particles. The particles were aligned using RELION 3D auto-refinement resulting in a consensus map with nominal resolution of 3.62 Å. Using the refinement parameters, subtractive 3D classification (N = 3) was performed on the particles by subtracting density outside of β′Si3 and classifying in a mask around β′Si3. Classification revealed three distinct β′Si3 dispositions (Figure S1D). Local refinement metadata (highlighted in red dotted box, Figure S1D) for TraR-Eσ$^{70}$(I) and TraR-Eσ$^{70}$(II) were used for RELION multi-body refinements to examine clamp motions (*Nakane et al., 2018*). Local resolution calculations were performed using blocres and blocfilt from the Bsoft package (*Cardone et al., 2013*).

## Acquisition and processing of Eσ$^{70}$ cryo-EM dataset

The Eσ$^{70}$ image acquisition and processing were the same as for TraR-Eσ$^{70}$ except with the following differences. Grids were imaged using a 200 keV Talos Arctica (FEI) equipped with a K2 Summit direct electron detector. Datasets were recorded with a pixel size of 1.3 Å over a defocus range of −1.0 μm to −2.5 μm. Movies were recorded in counting mode at 8.4 electrons/physical pixel/second in dose-fractionation mode with subframes of 0.2 s over a 10 s exposure (50 frames) to give a total dose of 84 electrons/physical pixel. Picked particles were extracted from the dose-weighted images in RELION (*Scheres, 2012*) using a box size of 200 pixels. The Eσ$^{70}$ dataset consisted of 3548 motion-corrected images with 1,387,166 particles. A subset of the particles was subjected to cryoSPARC ab-initio reconstruction (*Punjani et al., 2017*) to generate a 3D template for 3D classifications in cryoSPARC and 3D refinements in RELION (Scheres 2012). Particles were split into two groups (1st group: particles from images 1-2,000; 2nd group: particles from images 2001–3548. Particles from each group were curated using cryoSPARC heterogeneous refinement (N = 3) resulting in a subset of 479,601 particles for the first group and 329,293 particles for the second group. Curated particles were combined and a consensus refinement was performed in RELION using the cryoSPARC generated initial model resulting in a map with nominal resolution of 4.54 Å (without post-processing). Particles from this refinement (highlighted in red dotted box, *Figure 1—figure supplement 3*) were further analyzed using RELION multi-body refinement as described in the text (*Nakane et al., 2018*). Additionally, particles were further curated using RELION 3D classification (N = 3) without alignment. Classification revealed two lower resolution class and a higher resolution class. The higher resolution class containing 358,725 particles was RELION 3D auto-refined and subjected to RELION CTF refinement and RELION Bayesian Polishing (*Zivanov et al., 2018*). After polishing, particles were refined to a nominal resolution of 4.05 Å after RELION post-processing.

## Acquisition and processing of *rpsT* P2-RPo cryo-EM dataset

The *rpsT* P2-RPo cryo-EM image acquisition and processing were the same as for TraR-Eσ$^{70}$ except with the following differences. The imaging defocus range was 0.5 μm to 2.5 μm. Movies were recorded in super-resolution mode at eight electrons/physical pixel/second in dose-fractionation mode with subframes of 0.2 s over a 10 s exposure (50 frames) to give a total dose of 80 electrons/physical pixel. The *rpsT* P2-RPo dataset consisted of 6912 motion-corrected images with 973,481 particles. In RELION, a consensus refinement was performed using the extracted particles and a cryoSPARC generated initial model resulting in a 4.62 Å resolution map. Using the refinement parameters, 3D classification (N = 2) was performed on the particles without alignment. Classification

revealed a lower resolution class and a higher resolution class, the latter with 370,965 particles with nominal resolution of 4.38 Å after RELION 3D auto-refinement. Refinement metadata and post-processing were used as inputs for RELION CTF refinement and RELION Bayesian Polishing (*Zivanov et al., 2018*). Subsequent 3D classification (N = 3) was used to further classify the polished particles resulting in one junk class and two high-resolution classes (*Figure 2—figure supplement 1*). The highest resolution reconstruction (3.43 Å) contained 289,679 particles.

## Model building and refinement of cryo-EM structures

To build initial models of the protein components of the complexes, a crystal structure of *Eco* E$\sigma^{70}$ [PDB ID 4LJZ, with $\sigma^{70}_{1.1}$ from 4LK1; (*Bae et al., 2013*) was manually fit into the cryo-EM density maps using Chimera (*Pettersen et al., 2004*) and manually adjusted using Coot (*Emsley and Cowtan, 2004*). For TraR-E$\sigma^{70}$, $\sigma^{70}_{1.1}$ from 4LK1 (*Bae et al., 2013*) and TraR from 5W1S (*Molodtsov et al., 2018*) were also added. For *rpsT* P2-RPo, the promoter DNA was manually added. Appropriate domains of each complex were rigid-body refined, then subsequently refined with secondary structure and nucleic acid restraints using PHENIX real-space refinement (*Adams et al., 2010*).

## Purification of TraR and RNAP for transcription assays

IPTG (1 mM final) was used to induce expression of TraR (WT or variant) from *Eco* BL21 DE3 *dksA*:: Tn10 (RLG7075) host cells. TraR and variants were purified as described (*Gopalkrishnan et al., 2017*), either from His$_6$-TraR overexpression plasmids with removal of the His$_6$-tag with thrombin, or from His$_{10}$-SUMO-TraR constructs with removal of the His$_{10}$-SUMO-tag with Ulp1 protease, resulting in a 72 amino acid TraR lacking the N-terminal Met. WT-TraR purified by the two methods gave comparable results. WT and variant RNAPs were purified as described previously (*Ross et al., 2016*). The $\Delta 1.1\sigma^{70}$ was expressed and purified as described previously (*Chen et al., 2017*). E$\Delta 1.1\sigma^{70}$ was reconstituted with a 4:1 molar ratio of $\Delta 1.1\sigma^{70}$ to core RNAP. The purified core RNAP lacked detectable WT-$\sigma^{70}$ activity.

## In vitro transcription assays, site-directed mutagenesis, and TraR-RNAP binding assays

All of these procedures were carried out exactly as previously described (*Gopalkrishnan et al., 2017*).

## Details of flux calculator calculations

We modeled the hypothesized effects of RNAP mutants ($\Delta 1.1\sigma^{70}$) or TraR on the energy diagram for transcription initiation using the flux calculator developed by *Galburt (2018)*. Since the relevant rate constants for promoters inhibited or activated by TraR are not known, the energy diagrams illustrating the kinetic/energetic schemes for transcription initiation from a hypothetical inhibited and activated promoter (*Figure 4—figure supplement 1C*, *Figure 8*) are meant to only qualitatively illustrate features of the promoters and the effects of manipulations [either $\sigma^{70}_{1.1}$ deletion (*Figure 4—figure supplement 1C*) or addition of TraR (*Figure 8*)]. We hypothesize that TraR influences multiple steps along the RPo formation pathway so we wanted to model RPo formation as a three-step linear mechanism (*Buc and McClure, 1985*; *Roe et al., 1985*; *Sclavi et al., 2005*; *Rutherford et al., 2009*; *Ruff et al., 2015a*; *Hubin et al., 2017a*) rather than the two-step mechanism used in the flux calculator (*Galburt, 2018*). However, a kinetic analysis of ppGpp/DksA activation on the p*argI* promoter revealed only a very small (<2 fold) effect on $K_B$ (the equilibrium constant for the initial step of RNAP binding to the free promoter) and a large effect on subsequent isomerization rates (~16 fold; *Paul et al., 2005*). Similarly, activation by TraR at low and high (saturating) concentrations of RNAP gave the same fold activation on the p*thrABC* promoter, indicating TraR also does not affect the initial binding step. We make the assumption that deletion of $\sigma^{70}_{1.1}$ also would not dramatically affect the initial binding step. Therefore, the initial binding step (R + P $\rightleftharpoons$ RP1) was included in the illustrations of the kinetic scheme but did not need to be accounted for in the calculations of fold-inhibition or fold-activation because deletion of $\sigma^{70}_{1.1}$ or addition of TraR has no effect on this step. In this way, the rate constants for the first step of the flux calculator ($k_{on}$ and $k_{off}$; *Galburt, 2018*) could be used as the rate constants for the second step of our kinetic scheme

($k_2$ and $k_{-2}$; *Figure 4—figure supplement 1C*; *Figure 8*) and so on. The precise values used in the calculations are tabulated in *Supplementary file 2*. The illustrations (*Figure 4—figure supplement 1C*; *Figure 8*) were not taken directly from the flux calculator (*Galburt, 2018*) but have been skewed vertically to make the differences between the inhibited and activated promoter easier to visualize.

## Acknowledgements
We thank M Ebrahim and J Sotiris at The Rockefeller University Evelyn Gruss Lipper Cryo-electron Microscopy Resource Center for help with cryo-EM data collection, and A Feklistov, R Saecker, and other members of the Darst-Campbell Laboratory for helpful discussions. This work was supported by NIH grants R01 GM114450 to EAC, R01 GM37048 to RLG and R35 GM118130 to SAD.

## Additional information

### Funding

| Funder | Grant reference number | Author |
| --- | --- | --- |
| National Institutes of Health | R01 GM114450 | Elizabeth A Campbell |
| National Institutes of Health | R01 GM37048 | Richard L Gourse |
| National Institutes of Health | R35 GM118130 | Seth A Darst |

The funders had no role in study design, data collection and interpretation, or the decision to submit the work for publication.

### Author contributions
James Chen, Conceptualization, Investigation, Methodology; Saumya Gopalkrishnan, Courtney Chiu, Albert Y Chen, Investigation, Methodology; Elizabeth A Campbell, Conceptualization, Supervision, Funding acquisition, Methodology; Richard L Gourse, Conceptualization, Supervision, Funding acquisition; Wilma Ross, Conceptualization, Supervision, Methodology; Seth A Darst, Conceptualization, Supervision, Funding acquisition, Investigation, Methodology

### Author ORCIDs
James Chen (iD) http://orcid.org/0000-0002-2311-003X
Seth A Darst (iD) https://orcid.org/0000-0002-8241-3153

### Decision letter and Author response
Decision letter https://doi.org/10.7554/eLife.49375.sa1
Author response https://doi.org/10.7554/eLife.49375.sa2

## Additional files

### Supplementary files
• Supplementary file 1. Cryo-EM data acquisition and refinement parameters (*Chen et al., 2010*).
• Supplementary file 2. Details of flux calculator (*Galburt, 2018*) calculations.
• Supplementary file 3. RNAP conformational changes.
• Supplementary file 4. Plasmids.
• Supplementary file 5. Oligonucleotides and Geneblock sequences.
• Transparent reporting form

### Data availability
The cryo-EM density maps have been deposited in the EMDataBank under accession codes EMD-0348 [Eco TraR-Eσ70(I)], EMD-0349 [Eco TraR-Eσ70(II)], EMD-20231 [Eco TraR-Eσ70(III)], EMD-20230 (Eco Eσ70), EMD-20203 (rpsT P2-RPo), and EMD-20232 (rpsT P2-RPo2). The atomic coordinates

have been deposited in the Protein Data Bank under accession codes 6N57 [Eco TraR-Eσ70(I)], 6N58 [Eco TraR-Eσ70(II)], 6P1K (Eco Eσ70), and 6OUL (rpsT P2-RPo).

The following datasets were generated:

| Author(s) | Year | Dataset title | Dataset URL | Database and Identifier |
|---|---|---|---|---|
| Chen J, Chiu C, Campbell EA, Darst SA | 2019 | E. coli TraR-Esigma70(I) | https://www.emdataresource.org/EMD-0348 | EMDataResource, EMD-0348 |
| Chen J, Chiu C, Campbell EA, Darst SA | 2019 | E. coli TraR-Esigma70(II) | https://www.emdataresource.org/EMD-0349 | EMDataResource, EMD-0349 |
| Chen J, Chiu C, Campbell EA, Darst SA | 2019 | E. coli TraR-Esigma70(III) | https://www.emdataresource.org/EMD-20231 | EMDataResource, EMD-20231 |
| Chen J, Chiu C, Campbell EA, Darst SA | 2019 | E. coli Esigma70 | https://www.emdataresource.org/EMD-20230 | EMDataResource, EMD-20230 |
| Chen J, Chiu C, Campbell EA, Darst SA | 2019 | E. coli Esigma70-rpsT P2 RPo(I) | https://www.emdataresource.org/EMD-20203 | EMDataResource, EMD-20203 |
| Chen J, Chiu C, Campbell EA, Darst SA | 2019 | E. coli Esigma70-rpsT P2 RPo(II) | https://www.emdataresource.org/EMD-20232 | EMDataResource, EMD-20232 |
| Chen J, Chiu C, Campbell EA, Darst SA | 2019 | E. coli TraR-Esigma70(I) | http://www.rcsb.org/structure/6N57 | RCSB Protein Data Bank, 6N57 |
| Chen J, Chiu C, Campbell EA, Darst SA | 2019 | E. coli TraR-Esigma70(II) | http://www.rcsb.org/structure/6N58 | RCSB Protein Data Bank, 6N58 |
| Chen J, Chiu C, Campbell EA, Darst SA | 2019 | E. coli Esigma70 | http://www.rcsb.org/structure/6P1K | RCSB Protein Data Bank, 6P1K |
| Chen J, Chiu C, Campbell EA, Darst SA | 2019 | E. coli Esigma70-rpsT P2 RPo(I) | http://www.rcsb.org/structure/6OUL | RCSB Protein Data Bank, 6OUL |

The following previously published datasets were used:

| Author(s) | Year | Dataset title | Dataset URL | Database and Identifier |
|---|---|---|---|---|
| Bae B, Darst SA | 2013 | Crystal structure analysis of the E. coli holoenzyme | https://www.rcsb.org/structure/4LJZ | RCSB Protein Data Bank, 4LJZ |
| Bae B, Darst SA | 2013 | Crystal structure analysis of the E. coli holoenzyme | https://www.rcsb.org/structure/4lk1 | RCSB Protein Data Bank, 4LK1 |
| Murakami KS, Molodtsov V | 2017 | X-ray crystal structure of Escherichia coli RNA polymerase and TraR complex | https://www.rcsb.org/structure/5W1S | RCSB Protein Data Bank, 5W1S |
| Murakami KS | 2015 | X-ray crystal structur of Escherichia coli RNA polymerase sigma70 holoenzyme | https://www.rcsb.org/structure/4YG2 | RCSB Protein Data Bank, 4YG2 |

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
