## [Decision Letter]

**Acceptance summary:**

The current work presents a structural analysis of the transcription factor TraR bound to *E. coli* RNAP holoenzyme. TraR regulates transcription initiation in bacteria by binding directly to RNA polymerase (RNAP) as opposed to promoter DNA. The binding of TraR is shown to lead to significant and novel conformational changes in RNAP, which were not evident in prior crystallographic studies, that now account for TraR's function. The structural data, combined with biochemical analyses, provide a mechanism for how TraR binding alters the RNAP conformation and conformational dynamics to modulate the energy landscape of RNAP/promoter formation and transcription initiation. The findings represent a significant step toward enhancing our understanding of the molecular mechanism of how TraR and its homologs, such as DksA, regulate transcription.

**Decision letter after peer review:**

Thank you for submitting your article "*E. coli* TraR allosterically regulates transcription initiation by altering RNA polymerase conformation and dynamics" for consideration by *eLife*. Your article has been reviewed by two peer reviewers, and the evaluation has been overseen by a Reviewing Editor and Gisela Storz as the Senior Editor. The following individual involved in review of your submission has agreed to reveal their identity: Dong Wang (Reviewer #1).

The reviewers have discussed the reviews with one another and the Reviewing Editor has drafted this decision to help you prepare a revised submission.

Summary:

In the present manuscript, Darst and colleagues use cryo-electron microscopy to determine how TraR, a bacterial transcription factor related to the ppGpp-response factor DksA, regulates RNA polymerase function. Structures of *E. coli* RNAP and σ-70, with or without TraR, and of RNAP•σ-70 bound to a TraR-repressed promoter are presented. The authors find that upon binding of TraR, substantial conformational changes occur in RNAP that account for TraR's function. These changes are not seen in previous crystallographic analysis, which may have been limited by crystal packing constraints, and indicate that TraR modulates RNAP transcription initiation by binding and altering the conformation and dynamics of the polymerase. TraR-induced transitions are further shown by biochemical analyses to be critical for transcription activation or inhibition, depending on the kinetic features of regulated promoters. Taken together, a detailed mechanism is presented for how binding of TraR alters RNAP conformation and dynamics and to modulate the energy landscape of polymerase-promoter formation and transcription initiation.

The structural work is of significant interest, as the physical mechanisms through which TraR factor (and other transcription factors which bind directly to RNAP) regulate the kinetics of transcription initiation are poorly understood. The manuscript is clearly written and represents a nice example of the growing power of cryo-EM for the analysis of transcription complexes and their regulation. The work should be of general interest to scientists who work in transcription field; however, several issues need to be addressed before a decision on publication can be made.

Essential revisions:

1) One issue is that no structure of TraR bound to an initiation complex is provided. The authors mention in the Materials and methods that they were unable to obtain a DNA-bound complex of TraR-RNAP in the cryo-EM. This is unfortunate, as the conformational changes that occur in the presence of DNA will be important for fully defining the structural mechanism of TraR. The lack of a structure is puzzling, since it is shown that TraR is active under the cryo-EM conditions (Figure 1—figure supplement 1A). This raises some concerns that the cryo-EM conditions allow TraR-RNAP interactions and subsequent conformational changes, but that these changes are not congruent with those that are required to support DNA binding. In addition, there is a degree of speculation on this issue in the Discussion that does not seem fully substantied in lieu of a DNA-bound TraR-RNAP complex. Some commentary on this point is needed. It is also recommended that the authors tone down some of the language on this point.

2) Based on its blocking of the NTP-entry channel, a TraR-bound complex would be, by definition, a repressed complex for initiation. It is less clear how this fact impacts the interpretation of the kinked bridge helix as a mechanism of repression. A simpler model (related to the one laid out in the last paragraph of the Discussion) may be that activation/repression is dictated solely by the lifetime of RPo in the absence of NTPs. In a potentially interesting synthesis, could it be that the BH clash results in a very low affinity between TraR and RPo, which might also explain the inability to obtain TraR RPo structures? If so, it may be that forming TraR-RPc complexes with DNA at low temperature (4C) prior to freezing for cryo-EM would reveal closed complexes with TraR bound. Here, formation of RPo would have a large negative linkage with TraR binding and formation of RPo would eject TraR in much the way as it ejects σ1.1. This hypothesis could also directly tested by comparing the affinity of TraR to RNAP alone and preformed RNAP RPo with mismatched DNA.

3) In multiple instances, the manuscript suggests links between structural observations and kinetics or dynamics. However, these links are hypothetical, as no time-dependent data are presented; the observed conformational heterogeneity (which is well-characterized here) may not necessarily be truly dynamic. This mis-conception is strengthened by wording throughout the paper, exemplified by the following in the Introduction, "…new software tools allow for the analysis of molecular motions in the cryo-EM data…" and from the Abstract, "that quantitative analysis of RNAP conformational heterogeneity revealed TraR-induced changes in RNAP dynamics." While it is accepted that motions are implied by the heterogeneity, the language connecting these two phenomena should be more nuanced.

4) The authors state that, "dampening of motion extent is proposed to facilitate the "nucleation of strand opening". But the referenced manuscript (Feklistov, 2017) proposes that motions between cRNAP and oRNAP are important for strand opening, stating: "Thus, −11 base-pair perturbations obviate the requirement for cRNAP, confirming that transient clamp closure nucleates melting," and "−10 element recognition by cRNAP nucleates melting, followed by transition into oRNAP form, allowing DNA loading and unwinding in the cleft; the final closure seals the promoter complex." Based on the model presented in Figure 4 of Feklistov, 2017, it would seem that restricting the ability to form oRNAP may activate or inhibit strand separation depending on the kinetics of RP1-RP2-RPo. Please comment.

5) There is not a clear synthesis of the variety of inferred structural effects on initiation. How are the effects on the clamp, the bridge helix, and the SLIs integrated to produce activation or inhibition? A kinetic model/schematic that concretely summarizes the kinetic and structural hypotheses for the reader in the context of transitions between different intermediate structures would go a long way to clarifying and synthesizing the proposed TraR model.

[Editors' note: further revisions were requested prior to acceptance, as described below.]

Thank you for submitting your article "*E. coli* TraR allosterically regulates transcription initiation by altering RNA polymerase conformation" for consideration by *eLife*. Your article has been reviewed by two peer reviewers, and the evaluation has been overseen by a Reviewing Editor and Gisela Storz as the Senior Editor. The following individual involved in review of your submission has agreed to reveal their identity: Dong Wang (Reviewer #1).

The reviewers have discussed the reviews with one another and the Reviewing Editor has drafted this decision to help you prepare a revised submission.ß

Summary:

There are still a few questions surrounding the work, mostly with respect to the description of a mechanism for activation. These aspects of the manuscript require some attention.

Essential revisions:

1) It would seem from the response that the authors are preparing a subsequent manuscript that will directly address the issue of the mechanism bound to initiation complexes, and that this may be considerably more complicated than the effects of TraR on the polymerase alone. Given that the paper pertains to the mechanism of TraR-based regulation, it is important to delineate how the present work, in particular the molecular mechanisms in the Discussion, stands on its own in absence of the DNA-bound models. In particular, to what extent will the observations described here be relevant after the second paper comes out?

2) The response states that the mechanism of inhibition is due to the trapping of an early bound intermediate on repressed promoters. It is also mentioned that on these promoters, bound complexes are short lived and that TraR will likely be present for the entirety of the time. This seems sensible. However, it is also claimed that "Due to differences in the rate-determining steps at inhibited vs. activated promoters, again described in the Discussion, TraR facilitates rather than inhibits RPo formation at activated promoters. The longer lifetime of the open complex at these promoters results in dissociation of TraR before RNAP dissociates from the promoter, allowing NTP entry and catalysis to proceed." In addition, the response states that "In brief, on activated promoters with long RPo lifetime, TraR would be expected to dissociate from the promoter complex before RNAP would dissociate from the promoter, leaving ample time for activation to occur."

The proposed logic for activation is unclear. How can TraR activate transcription via early dissociation? What is "ample time for activation to occur?" After it leaves, one might expect that transcription would be de-repressed to basal levels – not activated. Also, relative to the landscapes shown in Figure 8: these are showing landscapes in the presence and absence of a factor, and factor dissociation during the process is not modeled. It is unclear how the landscapes relate to the hypothesis for activation. Please clarify these points in the manuscript.

3) The description of the mechanism via the use of energy landscapes has the potential to greatly clarify the hypotheses made and is a strong addition to the paper. The kinetic/thermodynamic features of the activated and repressed promoters are important should be described in the paper. It seems that one critical component may be the rate of escape. In addition, please provide details on how the flux calculator from Galburt et al. was adapted to a four-state scheme. The on-line calculator appears to only have three states.

4) The promoters shown as examples for activated-and-repressed (Figure 8), or as activated-but-no-effect (Figure 4—figure supplement 1) appear to differ in the rate of promoter escape, which is not mentioned in the manuscript. Could the effect of promoter escape rate revealed by the landscapes be the determinant for TraR-based activation/repression? The landscapes need to be described quantitatively so as to allow others to recapitulate the results. Please comment/clarify.

5) There is prior work on the role of different polymerase domains on initiation intermediates (Ruff et al., 2015) that should probably be considered.

---

## [Author Response]

Essential revisions:1) One issue is that no structure of TraR bound to an initiation complex is provided. The authors mention in the Materials and methods that they were unable to obtain a DNA-bound complex of TraR-RNAP in the cryo-EM. This is unfortunate, as the conformational changes that occur in the presence of DNA will be important for fully defining the structural mechanism of TraR. The lack of a structure is puzzling, since it is shown that TraR is active under the cryo-EM conditions (Figure 1—figure supplement 1A). This raises some concerns that the cryo-EM conditions allow TraR-RNAP interactions and subsequent conformational changes, but that these changes are not congruent with those that are required to support DNA binding. In addition, there is a degree of speculation on this issue in the Discussion that does not seem fully substantied in lieu of a DNA-bound TraR-RNAP complex. Some commentary on this point is needed. It is also recommended that the authors tone down some of the language on this point.

We were unable to form a TraR-RNAP-promoter DNA complex with an *rrnB* P1 promoter fragment, as mentioned in the Materials and methods. We have subsequently obtained cryo-EM structures of TraR-RNAP-promoter DNA complexes using the *rpsT* P2 promoter fragment (Figure 2A). Both *rrnB* P1 and *rpsT* P2 are inhibited by TraR. The *rrnB* P1 promoter is known to form a very unstable complex with RNAP with a very short half-life, which is further inhibited by TraR. The *rpsT* P2 promoter also forms relatively unstable complexes but more stable than *rrnB* P1 (Gopalkrishnan et al., 2017, Gourse et al., 2018), consistent with our results. Our study of the TraR-RNAP-*rpsT* P2 complexes has revealed 7 new structures that represent intermediates along the pathway of RPo formation – adequately addressing this work here is way beyond the scope of this manuscript and is the subject of a separate manuscript submitted elsewhere. Nevertheless, the cryo-EM conditions and subsequent conformational changes support DNA binding. Here we focus on conformational changes that TraR induces by binding to RNAP, which provide structure-based hypotheses for the differential effects of TraR on transcription initiation (activation of some promoters, inhibition of others). These hypotheses are then tested with biochemical experiments.

In response to this and other reviewer comments, we have completely re-written and re-organized the Discussion section.

2) Based on its blocking of the NTP-entry channel, a TraR-bound complex would be, by definition, a repressed complex for initiation. It is less clear how this fact impacts the interpretation of the kinked bridge helix as a mechanism of repression. A simpler model (related to the one laid out in the last paragraph of the Discussion) may be that activation/repression is dictated solely by the lifetime of RPo in the absence of NTPs. In a potentially interesting synthesis, could it be that the BH clash results in a very low affinity between TraR and RPo, which might also explain the inability to obtain TraR RPo structures? If so, it may be that forming TraR-RPc complexes with DNA at low temperature (4C) prior to freezing for cryo-EM would reveal closed complexes with TraR bound. Here, formation of RPo would have a large negative linkage with TraR binding and formation of RPo would eject TraR in much the way as it ejects σ1.1. This hypothesis could also directly tested by comparing the affinity of TraR to RNAP alone and preformed RNAP RPo with mismatched DNA.

The reviewer raises several interesting and important questions that are relevant to the complexity of the regulatory mechanism employed by TraR. We revised the Discussion, especially the sections entitled “Structural Mechanisms of Activation by TraR” and “Structural Mechanisms of Inhibition by TraR”, to address these questions and to describe our model more comprehensively. We also address these issues by pointing out specific data presented here, as well as in previous publications, that support our model, and we added a new figure (Figure 8) to summarize features of the model.

Specifically, we address the questions raised as follows:

2a) *‘Based on its blocking of the NTP-entry channel, a TraR-bound complex would be, by definition, a repressed complex for initiation.'*

Yes, we agree that a TraR-bound complex would be repressed for initiation, and this would be true whether the promoter is a negatively or positively regulated one. Thus, this cannot be the mechanism for activation. For the following reasons (also outlined in the Discussion) we do not think that blocking of NTP entry is the primary mechanism for inhibition.

Due to the TraR- induced conformational changes in RNAP (detailed in our paper) that severely reduce RPo formation by stabilizing an earlier intermediate on the pathway and by weakening duplex DNA interactions at inhibited promoters, transcription initiation is blocked before the stage of NTP entry and catalysis. Footprints of negatively-regulated promoter complexes containing TraR display protection from DNaseI characteristic of earlier intermediates rather than RPo, consistent with this interpretation.

Our earlier data (Lennon et al., 2009) showed that DksA (and by extension, its homolog TraR) has 10-fold reduced binding affinity for RPo relative to RNAP holoenzyme. Consistent with these data, recent single molecule fluorescence data from Gelles and colleagues (Stumper et al., 2019) showed that secondary channel binding factors (specifically GreB and DksA) bind to RNAP to form an RNAP-factor complex, before RNAP binds to the promoter, and the factors remain bound to RNAP on the promoter until RNAP dissociates. On negatively-regulated promoters like *rrnB* P1, which form extraordinarily short-lived promoter complexes with RNAP, the lifetime of the factor with RNAP is longer than the lifetime of RNAP with the promoter.

Due to differences in the rate-determining steps at inhibited vs. activated promoters, again described in the Discussion, TraR facilitates rather than inhibits RPo formation at activated promoters. The longer lifetime of the open complex at these promoters results in dissociation of TraR before RNAP dissociates from the promoter, allowing NTP entry and catalysis to proceed.

2b) *'It is less clear how this fact impacts the interpretation of the kinked bridge helix as a mechanism of repression. A simpler model (related to the one laid out in the last paragraph of the Discussion) may be that activation/repression is dictated solely by the lifetime of RPo in the absence of NTPs.'*

We agree with the reviewer that the lifetime of RPo is an important determinant of whether a promoter is activated or repressed, as described above, but we think that it is part of a more complex multistep mechanism. We propose that kinking of the bridge helix (BH) plays a role in the mechanism of inhibition by reducing the stability and occupancy of RPo by preventing a template strand interaction with the BH. In addition, the template strand-BH interaction correctly positions the template strand in the active site, and a BH kink would therefore inhibit catalysis. We suggest while most RNAP-TraR complexes at an inhibited promoter would not proceed to forming an open complex, the small fraction that do would be inhibited by the BH kink and restriction of NTP access, thus ensuring efficient inhibition.

2c) *‘In a potentially interesting synthesis, could it be that the BH clash results in a very low affinity between TraR and RPo, which might also explain the inability to obtain TraR RPo structures?'*

Yes, this idea is supported by our previous observation of a 10-fold reduced affinity of DksA, a TraR homolog, for a pre-formed RPo (Lennon et al., 2009). Since TraR has a similar affinity for RNAP (Gopalkrishnan et al., 2017), we propose that TraR also has a reduced affinity for pre-formed RPo. We suggest that RPo formation can occur in a small fraction of complexes in the presence of bound TraR, and that blocking catalysis would be part of a multipartite mechanism that would ensure efficient inhibition of this small fraction of complexes.

2d) *'If so, it may be that forming TraR-RPc complexes with DNA at low temperature (4C) prior to freezing for cryo-EM would reveal closed complexes with TraR bound. Here, formation of RPo would have a large negative linkage with TraR binding and formation of RPo would eject TraR in much the way as it ejects σ1.1. This hypothesis could also directly tested by comparing the affinity of TraR to RNAP alone and preformed RNAP RPo with mismatched DNA.'*

Several different TraR-RNAP-promoter complexes formed with DNA templates containing or lacking mismatches have been analyzed using single particle cryo EM. These results identify the structures of several different intermediates in the RPo formation pathway. Their description is beyond the scope of the present manuscript and is being reported elsewhere.

3) In multiple instances, the manuscript suggests links between structural observations and kinetics or dynamics. However, these links are hypothetical, as no time-dependent data are presented; the observed conformational heterogeneity (which is well-characterized here) may not necessarily be truly dynamic. This mis-conception is strengthened by wording throughout the paper, exemplified by the following in the Introduction, "…new software tools allow for the analysis of molecular motions in the cryo-EM data…" and from the Abstract, "that quantitative analysis of RNAP conformational heterogeneity revealed TraR-induced changes in RNAP dynamics." While it is accepted that motions are implied by the heterogeneity, the language connecting these two phenomena should be more nuanced.

There are two types of conformational heterogeneity characterized by cryo-EM in this study – we can call them digital and analog:

I) Digital – The complexes exist in relatively discrete conformational states that can be parsed using the software tools in RELION. These are represented here by TraR-Eσ^70^(I) and TraR-Eσ^70^(II), where the two consensus structures differ in the disposition of β'Si3, which is seen in two discrete positions (Figures 1A, B).

II) Analog – Within these classes of particles [TraR-Eσ^70^(I) and TraR-Eσ^70^(II)], the RNAP clamp is also conformationally heterogeneous but does not parse into discrete bins. Rather, the clamp positions are characterized by continuous ~Gaussian distributions (Figure 6B-D).

A 'digital' conformational heterogeneity would be characterized by lifetimes of the discrete states that are long compared to the freezing time of the sample on the grid (unknown, but probably ~10 μs), while the 'analog' conformational heterogeneity would have lifetimes of any particular conformation short compared to the freezing time. Although both types of conformational heterogeneity are almost certainly dynamic on some time scale, the reviewers comments appear to relate to the 'analog' conformational heterogeneity observed for the RNAP clamp. It seems that the reviewer is suggesting that the continuous ~Gaussian distributions of RNAP clamp positions observed here in the frozen cryo-EM samples do not represent dynamic motions in solution that were trapped during the freezing process, but instead represent static conformational states; this seems inconceivable to us. This is supported by the finding of dynamic clamp behavior by single-molecule FRET (Duchi et al., 2018), which we have noted now in the manuscript (subsection “TraR binding restricts the range of clamp positions in Eσ^70^”, first paragraph). Nevertheless, we have removed mentions of clamp 'dynamics' or 'motions' and replaced these with 'conformational heterogeneity' or other wording, including in the Title, in the Abstract, as well as in the Results, and in the (rewritten) Discussion.

4) The authors state that, "dampening of motion extent is proposed to facilitate the "nucleation of strand opening". But the referenced manuscript (Feklistov, 2017) proposes that motions between cRNAP and oRNAP are important for strand opening, stating: "Thus, −11 base-pair perturbations obviate the requirement for cRNAP, confirming that transient clamp closure nucleates melting," and "−10 element recognition by cRNAP nucleates melting, followed by transition into oRNAP form, allowing DNA loading and unwinding in the cleft; the final closure seals the promoter complex." Based on the model presented in Figure 4 of Feklistov, 2017, it would seem that restricting the ability to form oRNAP may activate or inhibit strand separation depending on the kinetics of RP1-RP2-RPo. Please comment.

By 'nucleation of strand opening', we mean initial melting of the -10 element (as detected by the RNAP beacon assay used in Feklistov et al., 2017). This paper showed that reagents that close the clamp (i.e. switch region antibiotics such as myxopyronin or T7 Gp2) significantly increase the rate of transcription bubble nucleation. We propose here that restricting the range of clamp opening by TraR could also have this effect (facilitate the very initial steps of -10 element melting). We have tried to clarify this statement (subsection “Conclusion”).

The reviewer is correct that Feklistov et al., 2017, also suggested that later steps of RPo formation would require clamp opening, so TraR restriction of clamp opening would be expected to hinder later steps, but this would not be inconsistent with our model – it would depend on the importance of the later steps to the overall kinetics of RPo formation. Note also that potential hindrance by TraR of later steps due to restrictions in clamp opening would be layered on top of the activation of later steps due to TraR's effect on facilitating σ^70^_1.1_ release (Figure 4D).

Also please note that modulation of clamp conformations is only a small part of the multipartite TraR mechanism (Figure 8).

5) There is not a clear synthesis of the variety of inferred structural effects on initiation. How are the effects on the clamp, the bridge helix, and the SLIs integrated to produce activation or inhibition? A kinetic model/schematic that concretely summarizes the kinetic and structural hypotheses for the reader in the context of transitions between different intermediate structures would go a long way to clarifying and synthesizing the proposed TraR model.

We have added such a figure (Figure 8) by adapting the web-based flux calculator (Galburt, 2018) to a four-step linear mechanism for transcription initiation. The figure shows hypothetical energy landscapes for an inhibited and an activated promoter. The figure illustrates proposed effects of TraR binding on specific features of the energy landscapes and shows how TraR can inhibit some promoters but activate others.

[Editors' note: further revisions were requested prior to acceptance, as described below.]

Essential revisions:1) It would seem from the response that the authors are preparing a subsequent manuscript that will directly address the issue of the mechanism bound to initiation complexes, and that this may be considerably more complicated than the effects of TraR on the polymerase alone. Given that the paper pertains to the mechanism of TraR-based regulation, it is important to delineate how the present work, in particular the molecular mechanisms in the Discussion, stands on its own in absence of the DNA-bound models. In particular, to what extent will the observations described here be relevant after the second paper comes out?

The current paper focuses on the structural mechanisms for both inhibition and activation by TraR. The second paper uses TraR as a tool to populate promoter melting intermediates with an inhibited promoter. The subject of this second paper is not TraR, nor its mechanism. The second paper describes the structures of these intermediates and also presents structure-based biochemical tests to support our proposal that these intermediates describe the normal promoter melting pathway even in the absence of TraR. Thus, the subject of the second paper is promoter melting, not TraR mechanism. There are no results in the second paper that contradict or supersede the results of this *eLife* submission.

2) The response states that the mechanism of inhibition is due to the trapping of an early bound intermediate on repressed promoters. It is also mentioned that on these promoters, bound complexes are short lived and that TraR will likely be present for the entirety of the time. This seems sensible. However, it is also claimed that "Due to differences in the rate-determining steps at inhibited vs. activated promoters, again described in the Discussion, TraR facilitates rather than inhibits RPo formation at activated promoters. The longer lifetime of the open complex at these promoters results in dissociation of TraR before RNAP dissociates from the promoter, allowing NTP entry and catalysis to proceed." In addition, the response states that "In brief, on activated promoters with long RPo lifetime, TraR would be expected to dissociate from the promoter complex before RNAP would dissociate from the promoter, leaving ample time for activation to occur."The proposed logic for activation is unclear. How can TraR activate transcription via early dissociation? What is "ample time for activation to occur?" After it leaves, one might expect that transcription would be de-repressed to basal levels – not activated. Also, relative to the landscapes shown in Figure 8: these are showing landscapes in the presence and absence of a factor, and factor dissociation during the process is not modeled. It is unclear how the landscapes relate to the hypothesis for activation. Please clarify these points in the manuscript.

We have further revised the Discussion section on “Structural mechanism for TraR-mediated activation” to clarify the model and address the questions raised by the reviewer. The first question concerned how dissociation of TraR from the open complex (RPo) could activate transcription. Confusion about this subject could have arisen because we previously addressed it at the end of the Discussion section on “Structural Mechanism of TraR-mediated inhibition” instead of in the section on “Structural mechanism of activation”. We have now moved the explanation to the activation section so that all features of the proposed model for activation are now presented together in one section.

In summary, we have previously published data showing that TraR and its homolog DksA enhance the transcription output from activated promoters relative to that with RNAP alone (Paul et al., 2005; Gopalkrishnan et al., 2017), and that RPo at activated promoters is very stable to competitor challenge (Barker et al., 2001). In addition, kinetic analyses showed that the presence of DksA enhanced the rate of formation and the amount of competitor stable RPo formed at an activated promoter relative to that formed with RNAP alone (Paul et al., 2005). In the present paper we present structural and biochemical evidence that these effects occur primarily because TraR-induces conformational changes in the β subunit that allosterically alter the position of σ_1.1_ in the main DNA binding channel (Figures 1, 4; Video 1), thereby increasing the ability of promoter DNA to displace σ_1.1_ from this channel. TraR also alters the position of β’Si3, contributing to activation (Figure 3). Contrary to what is implied in the comment, these enhancements to the overall rate occur while TraR is still present in the complex – the increase in the rate and amount of RPo formed while TraR was present leads to the increase in transcription once TraR dissociates.

Since the structural data shown here indicate that TraR bound to the complex would sterically block iNTP access to the active site, t-strand positioning and catalysis (Figure 5; see also Molodtsov et al., 2018), TraR must therefore dissociate from the complex in order for transcription to initiate from the activated promoter. The proposal that TraR dissociates from RPo at an activated promoter prior to dissociation of RNAP from the promoter is consistent with previously published data. The lifetimes of RPo at activated promoters (measured in hours; Barker et al., 2001) are much longer than that of the transcription factors on RNAP (GreB, DksA and by extension TraR; measured in seconds; Stumper et al., 2019).

A second part of the reviewer’s comments concerns whether TraR “de-represses basal activity” rather than really activates transcription. We refer to “basal” activity as the activity observed with RNAP alone in the absence of TraR. We have shown previously that TraR increases transcription 4-6 fold over the amount observed without TraR, i.e. it really activates (Gopalkrishnan et al., 2017; also see Figures 3E, 3G, Figure 3—figure supplement 1B, C, E, G, Figure 4C, D). We have revised a paragraph (subsection “A TraR-induced ~18° rotation of βlobe-Si1 plays a major role in transcription regulation”, fourth paragraph; re Figure 4D) to clarify what is meant by basal transcription.

We have also included a description in the revised Discussion section about how the energy landscape shows the activation step. We have also modified the kinetic scheme at the top of Figure 8 to reflect TraR (T) dissociation that is required for initiation of RNA chain synthesis and transition of RPo to RPitc.

3) The description of the mechanism via the use of energy landscapes has the potential to greatly clarify the hypotheses made and is a strong addition to the paper. The kinetic/thermodynamic features of the activated and repressed promoters are important should be described in the paper. It seems that one critical component may be the rate of escape. In addition, please provide details on how the flux calculator from Galburt et al. was adapted to a four-state scheme. The on-line calculator appears to only have three states.

The general kinetic/thermodynamic features of inhibited vs. activated promoters are discussed in the Discussion. When the energy diagrams are first introduced (subsection “A TraR-induced ~18° rotation of βlobe-Si1 plays a major role in transcription regulation”, last paragraph), we have added a note that the features of inhibited vs. activated promoters are described later in the Discussion. We have also added new references establishing the 3-step minimal linear mechanism for RPo formation (or 4-step mechanism if one includes promoter escape). Basically, complexes at inhibited promoters pass through the pathway rapidly but the final RPo is relatively unstable; in the absence of high NTP concentrations, RPo exists in equilibrium with earlier intermediates. By contrast, activated promoters form RPo slowly, but the RPo that is formed is very stable. In terms of the basic features of the energy diagram, inhibited promoters have low kinetic barriers (low peaks between the valleys) but the valleys going from left to right are generally uphill. Activated promoters have high kinetic barriers (high peaks between the valleys) but the valleys going from left to right are generally downhill. We have tried to capture these basic features in the basal energy diagrams (the black lines) in Figure 4—figure supplement 1C and Figure 8 (these figures have been revised to better illustrate these points).

Regarding the rate of escape, there is no evidence that promoter escape is rate-limiting or that TraR alters the rate of escape from either the activated or inhibited promoters used here. Our models for activation and inhibition propose that TraR would not be associated with the final RPo at the activated promoters (see above response to comment 2), so would not affect the escape rate. For inhibited promoters, TraR depopulates RPo by stabilizing an earlier intermediate complex, thereby preventing initiation. Thus, it is unlikely that TraR remains bound to RPo except in a small fraction of the population in which TraR binding would be expected to sterically occlude catalysis (Figure 5; Molodtsov et al., 2018).

The details of how we used the flux calculator of Galburt, 2018, are now described in Materials and methods ('Details of flux calculator calculations'). The input parameters and results of the flux calculator calculations are now tabulated in Supplementary file 2.

4) The promoters shown as examples for activated-and-repressed (Figure 8), or as activated-but-no-effect (Figure 4—figure supplement 1) appear to differ in the rate of promoter escape, which is not mentioned in the manuscript. Could the effect of promoter escape rate revealed by the landscapes be the determinant for TraR-based activation/repression? The landscapes need to be described quantitatively so as to allow others to recapitulate the results. Please comment/clarify.

The details of how we used the flux calculator of Galburt, 2018, are now described in Materials and methods ('Details of flux calculator calculations'). The input parameters and results of the flux calculator calculations are now tabulated in Supplementary file 2. The rates of escape for the inhibited promoter in Figure 4—figure supplement 1C and in Figure 8, and the activated promoter in the two cases, are the same, and these are not altered by TraR in our model.

5) There is prior work on the role of different polymerase domains on initiation intermediates (Ruff et al., 2015) that should probably be considered.

We agree with the reviewer that previous work about the role of different RNAP domains should be mentioned. We have included a sentence (subsection “Structural mechanism for TraR-mediated activation”, second paragraph) describing conclusions from previous work from the Record and Hinton labs on effects of the deletion of σ_1.1_ on promoter complex formation and stability. Their results, specifically that deletion of σ_1.1_ increases RPo formation at some promoters and decreases it at others, are consistent with effects of the σ_1.1_ deletion on transcription observed here at activated or inhibited promoters (Figure 4; Figure 4—figure supplement 1). Work from the Record lab also showed a role for Si3 in stabilization of RPo (Ruff et al., 2015b). This is consistent with our finding that Si3 plays a role in promoter activation by TraR (Figure 3) and suggests that Si3 could affect activation by altering complex stability. This is now mentioned in our revised Discussion section (see the fourth paragraph of the aforementioned subsection).